# Capability Localization: Capabilities Can be Localized rather than Individual Knowledge

**Xiusheng Huang**[* 1,2,3]**, Jiaxiang Liu**[* 1,2]**, Yequan Wang**[† 3]**, Jun Zhao**[1,2] **and Kang Liu**[† 1,2]
[1]The Key Laboratory of Cognition and Decision Intelligence for Complex Systems,
Institute of Automation, Chinese Academy of Sciences
[2]School of Artificial Intelligence, University of Chinese Academy of Sciences
[3]Beijing Academy of Artificial Intelligence, Beijing, China
`huangxiusheng2020@ia.ac.cn, liujiaxiang21@mails.ucas.ac.cn,`
`tshwangyequan@gmail.com, {jzhao,kliu}@nlpr.ia.ac.cn`

## Abstract

Large scale language models have achieved superior performance in tasks related to natural language processing, however, it is still unclear how model parameters affect performance improvement. Previous studies assumed that individual knowledge is stored in local parameters, and the storage form of individual knowledge is dispersed parameters, parameter layers, or parameter chains, which are not unified. We found through fidelity and reliability evaluation experiments that individual knowledge cannot be localized. Afterwards, we constructed a dataset for decoupling experiments and discovered the potential for localizing data commonalities. To further reveal this phenomenon, this paper proposes a **C**ommonality **N**euron **L**ocalization (**CNL**) method, which successfully locates commonality neurons and achieves a neuron overlap rate of 96.42% on the GSM8K dataset. Finally, we have demonstrated through cross data experiments that commonality neurons are a collection of capability neurons that possess the capability to enhance performance. Our code is available at https://github.com/nlpkeg/Capability-Neuron-Localization.

## 1 Introduction

Large scale language models (LLMs) have received widespread attention due to their superior performance in the field of natural language processing (Zhao et al., 2023; Yao et al., 2023). Although LLMs have demonstrated extraordinary capabilities, humans are still unclear about the relationship between model parameters and superior performance (MacAskill & Kittler, 2010). More and more research is focusing on the security (Bonaldi et al., 2024; Sun et al., 2024), ethics (Yan et al., 2024; Haltaufderheide & Ranisch, 2024), and potential performance of models, with the black box nature of models being the main limitation of these studies (Guidotti et al., 2018; Huang et al., 2025). Therefore, establishing a mapping relationship between the internal parameters and capabilities of the model is becoming increasingly important (Ding et al., 2023; Huang et al., 2024).

Recent research has mainly focused on the correspondence between individual knowledge and parameters (Ledeen et al., 1976). Specifically, KN (Dai et al., 2021) believes that individual knowledge is stored on distributed parameters, and the validity of the conclusions is verified by increasing or zero the activation. ROME (Meng et al., 2022a) believes that individual knowledge is stored on the entire parameter layer, and then utilizes knowledge editing techniques to prove that modifying the parameters of the entire layer can modify individual knowledge. Finally, the KC (Yao et al., 2024) assumes that individual knowledge is stored in a parameter chain and utilizes the entire parameter chain to recall the knowledge. Different jobs believe that individual knowledge has different storage forms, which has caused the following difficulties for researchers: (A) What is the storage form of individual knowledge? (B) If the existing locating methods are inaccurate, can individual knowledge really be parameter localized?

---

[*]Equal contribution.
[†]Corresponding authors.

To address the above challenges, we design experiments for evaluating fidelity and reliability (Veh et al., 1995). Specifically, for the fidelity experiment, we assume that utilizing the rewritten individual knowledge prompt, the similarity of the located neurons should have a higher degree of overlap compared to the original (Trimmer, 2015). Therefore, we constructed a knowledge rewriting dataset to evaluate the fidelity of previous knowledge localization methods on individual knowledge. The experimental results show that the coincidence degree of causal tracing (Meng et al., 2022a) in ROME is the highest, reaching 37.3%, while the coincidence degrees of KN (Dai et al., 2021) and KC (Yao et al., 2024) are 27.5% and 32.7%, respectively. This indicates that previous knowledge localization methods were not faithful to the same individual knowledge. The answer to question A is: **The existing forms of individual knowledge storage are all inaccurate**.

Previous knowledge localization methods have proposed corresponding validation methods, reliability experiments will evaluate the reliability of these methods. Firstly, to verify that individual knowledge is stored on distributed parameter, KN (Dai et al., 2021) verifies the correspondence between the individual knowledge and the parameters by increasing or zeroing the activation. Our reliability experiments indicate that increasing or zeroing activation does not necessarily lead to an enhancement or weakening of corresponding knowledge. At the same time, we utilize corresponding models to demonstrate that there is no strong correlation between the parameters and knowledge. Secondly, to verify that individual knowledge is stored on the parameter layer, ROME (Meng et al., 2022a) edits the knowledge by updating the parameters of the entire layer (Peng et al., 2024; Liu et al., 2024). However, reliability experiments can still achieve knowledge editing by editing other layers (not localized layers). Finally, to verify that individual knowledge is stored on the parameter chains, KC (Yao et al., 2024) utilizes the localized parameter chains to recall the knowledge. The reliability experiment proves that the accuracy of using the parameter chain recall knowledge for localization (top k=1) is 12.3%. In addition, the entire parameters chain occupies 2.6% of the overall model parameters, and the granularity of knowledge and parameters does not match, which does not indicate the correspondence between parameters and knowledge. Using reliability experiments, the answer to question B is: **Existing technologies cannot localize individual knowledge parameters**.

To further reveal the form of knowledge storage, we designed 1000 comparative samples and conducted decoupling experiments. For example, the comparative samples include subsample 1 "providing the correct answer for 1+1=?" and subsample 2 "programming as Python code for 1+1=?". This comparative sample contains the same main part "1+1=?", which also tests the mathematical and programming capabilities of the model. Utilizing gradient response analysis (Hinterstoisser et al., 2011), it is found that the coincidence rate of response parameters between subsample 1 and subsample 2 was 15.6%. Interestingly, we find that the coincidence rate of response parameters for all subsample 1 was 7.3%, which is 8.6% for all subsample 2. **Individual knowledge cannot achieve parameter localization, can the commonality of data be achieved?**

To verify this hypothesis, we propose a Commonality Neurons Locating (CNL) method. We utilize samples of the same type to obtain commonalities in the data and successfully located commonality neurons. Furthermore, we demonstrate through cross datasets experiments that **commonality neurons are a collection of capability neurons that possess the capability to enhance performance.**

To our knowledge, we are the first to prove the unreliability of existing individual knowledge localization conclusions and point out that the capability can achieve local parameterization, which will help reveal the utility of internal parameters in the model. Our contributions can be summarized as follows:

- In order to clarify the storage form of individual knowledge, we conduct fidelity and reliability evaluation experiments. The experimental results indicate that existing localization methods are not faithful to individual knowledge. At the same time, existing validation methods cannot support parameter localization with individual knowledge.

- To further reveal the form of knowledge storage, we conduct decoupling experiments. The experiment find that the coincidence rate of response parameters between subsamples with the same main part is only 15.6%, while the coincidence rate of response parameters between samples that test the same capability of the model is 8.6%, which means that the commonality of data corresponds to parameters.

- We propose a Commonality Neurons Locating method and successfully located commonality neurons. Furthermore, we demonstrate through cross datasets experiments that commonality

neurons are a collection of capability neurons, and establish a mapping relationship between capabilities and parameters.

## 2 BACKGROUND

Recent research has mainly focused on the correspondence between individual knowledge and parameters, and suggests that the storage forms of individual knowledge are distributed parameters, parameter layers, or parameter chains. Next, we will introduce three locating methods and their corresponding validation experiments.

### 2.1 DISTRIBUTED PARAMETERS

**Locating Method.**  Given an input prompt $x = [x_1, \cdots, x_X]$, we first define the model output $P_{x,y^*}(\widehat{\omega_X^{l,j}})$ as the probability of the correct answer predicted by a pretrained model:

$$P_{x,y^*}(\widehat{\omega_X^{l,j}}) = p(y^*|x, \omega_X^{l,j}[x] = \widehat{\omega_X^{l,j}}) \tag{1}$$

where $y^*$ denotes the correct answer, $\widehat{\omega_X^{l,j}}$ is defined in the Appendix B and is a given constant that the output $\omega_X^{l,j}[x]$ is assigned to. In order to calculate the attribution score of a neuron $Attr(\omega^{l,j})$, we gradually change $\omega_X^{l,j}[x]$ from 0 its original value $\overline{\omega_X^{l,j}[x]}$ calculated by the pretrained model, and meanwhile integrate the gradients:

$$Attr(\omega^{l,j}) = \overline{\omega_X^{l,j}[x]} \int_{\alpha=0}^1 \frac{\partial P_{x,y^*}(\alpha \overline{\omega_X^{l,j}[x]})}{\partial \omega_X^{l,j}[x]} d\alpha \tag{2}$$

where $\frac{\partial P_{x,y^*}(\alpha \overline{\omega_X^{l,j}[x]})}{\partial \omega_X^{l,j}[x]}$ calculates the gradient of the model output with regard to $\omega^{l,j}$. As $\alpha$ changes from 0 to 1, by integrating the gradients, $Attr(\omega^{l,j})$ accumulates the output probability change caused by the change of $\omega_X^{l,j}[x]$. If the neuron has a great influence on the expression of a fact, the gradient will be salient, which in turn has large integration values. Therefore, the attribution score can measure the contribution of the neuron $\omega^{l,j}$ to the factual expressions.

**Verification Experiment.**  Given a relational fact, we manipulate its knowledge neurons in two ways: (1) suppressing knowledge neurons by setting their activations to 0; (2) amplifying knowledge neurons by doubling their activations.

### 2.2 PARAMETER LAYERS

**Locating Method.**  Similar to the causal tracing (Meng et al., 2022b), a clean run that predicts the fact, a corrupted run where the prediction is damaged, and a corrupted-with-restoration run that tests the capability of a single state to restore the prediction. More details are displayed in Appendix C.

**Verification Experiment.**  This method assumes that individual knowledge is stored in the parameter layer and validated using knowledge editing techniques. Specifically, by updating the entire parameter layer, the original knowledge of the model is changed and assumed to be stored in that layer.

### 2.3 PARAMETER CHAINS

**Locating Method.**  KC (Yao et al., 2024) believes that individual knowledge is stored on a parameter chain and utilizes the entire parameter chain to recall knowledge. More details are shown in Appendix D.

If the score $S(e_i)$ is less than the predefined threshold, the $e_i$ refers to calculating the edges in graph G, they consider the edge to be non-critical and remove it from the computation graph, updating the

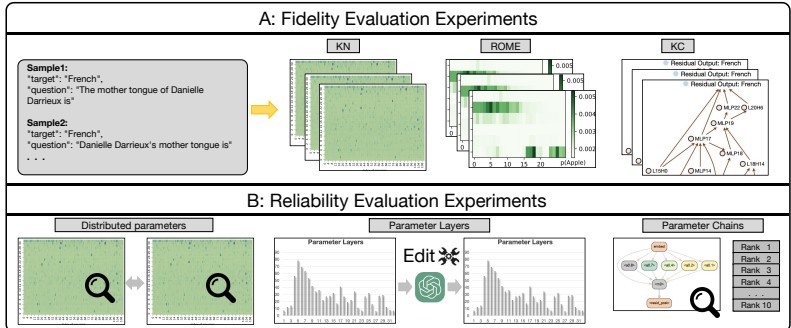

Fig. 1: Overall framework diagram of fidelity and reliability evaluation experiments. Experiment A represents the visualization results of samples with the same semantic localization, while experiment B represents the impact of operating localization neurons on performance

temporary circuit. They first sort the graph by topological rank and traverse all edges in this manner, they derive a circuit $C_k$ that contributes to representing the knowledge necessary to answer the factual question:

$$C_k =< N_k, E_k >$$ (3)

where $C_k$ is the circuit for the knowledge triplet k, consisting of the nodes $N_k$ and edges $E_k$.

**Verification Experiment.** To verify that individual knowledge is stored on the knowledge chain, KC (Yao et al., 2024) utilizes the parameter chain of localization to recall the knowledge. They measure the rank of the target entity o among the top 10 predicted tokens.

## 3 EVALUATE INDIVIDUAL KNOWLEDGE LOCALIZATION METHODS

We will introduce fidelity and reliability evaluation experiments in this section.

### 3.1 FIDELITY EVALUATION EXPERIMENTS

As show in Fig 1, the fidelity experiment is mainly aimed at verifying the fidelity of existing individual knowledge localization methods to individual knowledge. We observe the overlap score of each knowledge localization result by rewriting a individual knowledge prompt 5 times. The following experiments are uniformly set as follows: base model is GPTJ (Achiam et al., 2023) model.

**Rewritten Dataset.** We first randomly select 1000 factual samples from the COUNTERFACT (Meng et al., 2022a) dataset, and rewrite the samples 4 times using GPT4o (Achiam et al., 2023). We will obtain 5 samples $(D|x_{ij}, i \in [1, 2, ...1000], j \in [1, 2, ...5])$ with the same semantic meaning.

**Distributed Parameters.** Kn (Dai et al., 2021) believes that individual knowledge is stored on a few parameters, with an average of 15.3 localized parameters found per sample. We will match the localization results of 5 samples $(x_{ij}, j \in [1, 2, ...5])$ with the same semantic meaning pairwise. As a result, it is found that the overlap of localization parameters for samples with the same semantic meaning was only **37.3%**, which clearly does not meet the expectation of locating individual knowledge.

**Parameter Layers.** ROME (Meng et al., 2022a) believes that individual knowledge is stored on the parameter layer, and we select the top k=3 parameter layer as the localization result. We will match the localization results of 5 samples with the same semantic meaning pairwise. As a result, it is found that the overlap of localization parameters for samples with the same semantic meaning was only **32.7%**. At the same time, the parameter layer accounted for a large proportion of the entire model, and the granularity of individual knowledge and parameters did not match.

**Parameter Chains.** KC (Yao et al., 2024) believes that individual knowledge is stored on a parameter chain, and we compare the parameter chains of sample localization with the same semantic meaning. Determine the fidelity of the locating method based on whether the parameter chains are the same. The experimental results show that the coincidence degree of the positioning parameter chain obtained using KC (Yao et al., 2024) is only **7.2%**. In addition, the average proportion of parameters in the parameter chain to the entire model is **1.6%**, and the granularity of individual knowledge and parameters does not match.

Three fidelity experiments indicate that existing single knowledge localization methods are not faithful to knowledge. Samples with the same semantic meaning cannot obtain identical or highly similar localization results using localization knowledge.

### 3.2 RELIABILITY EVALUATION EXPERIMENTS

As show in Fig 1, the existing single knowledge localization methods all have separate validation methods. Below we will introduce reliability experiments for locating methods. The following experiments are uniformly set as follows: base model is GPTJ (Achiam et al., 2023) model, dataset is factual dataset zsRE (Levy et al., 2017).

**Distributed Parameters.** KN (Dai et al., 2021) verifies the effectiveness of the localization parameters by setting the activation value to zero or increasing it. We randomly select equal amounts of parameters, set them to zero or increase the activation value, and compare the experimental results with the located parameters.

The experimental results show that when we double the localized activation values, 85.1% of samples increase their target probabilities, while **14.9%** decrease. For random activation values, 72.3% of samples see an increase, and **27.7%** a decrease. Additionally, when we set localized activation values to zero, **27.4%** of samples increase their target probabilities, while 72.4% decrease. For random activation values, **17.6%** of samples increase, and 82.6% decrease. The experimental results are similar to the random activation values, which cannot fully demonstrate the effectiveness of the verification method.

**Parameter Layers.** ROME (Meng et al., 2022a) utilizes knowledge editing techniques to verify the effectiveness of the parameter layer. ROME chooses to edit the parameters of the 5-th layer for validation, and utilizes the changes in the model's output as a basis to prove that a single knowledge exists in a single layer. In Fig 2, we randomly edited all layers and found that editing other layers resulted in similar performance, indicating that the parameter layer validation method is ineffective.

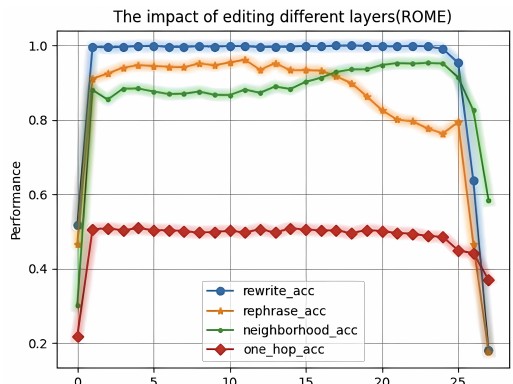

Fig. 2: The performance of editing at different layers, the horizontal axis is layer number.

**Parameter Chains.** KC (Yao et al., 2024) utilizes the parameter chain recall knowledge after localization to verify the effectiveness of localization. We utilize the parameter chain recall knowledge after localization to select the candidate items with rank=1 that are equal to the target samples as successful samples. The experimental results showed that the success rate of recall knowledge was only **12.3%**. At the same time, we find that the samples with successful recall contains a large number of parameters, accounting for an average of **2.6%** of the entire model parameters.

## 4 DECOUPLING EXPERIMENT

By evaluating previous knowledge localization methods, we find that none of the existing methods are convincing. In order to further reveal the form of knowledge storage, we design decoupling experiments to decouple different factors of knowledge and explore the corresponding relationships between parameters and indicators.

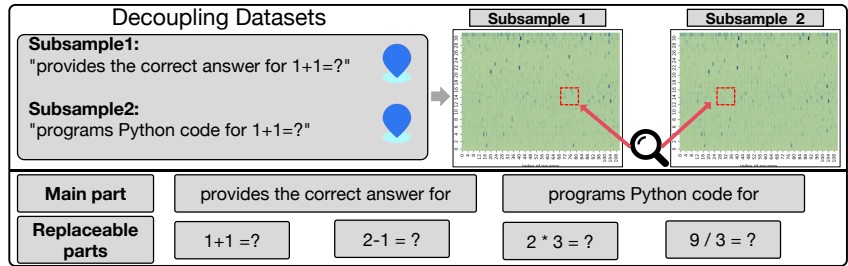

Fig. 3: Overall framework diagram of decoupling experiments. The upper half represents the overlapping neurons for obtaining the localization of subsample1 and subsample2, while the lower half represents the composition of the subsample, consisting of the main and replaceable parts.

## 4.1 DECOUPLING DATASETS

We design 1000 comparative samples and conduct decoupling experiments. Each comparison sample contains two sub samples, whose replaceable parts remain consistent but differ in main parts. This helps us better analyze the correspondence between the parameters and indicators within the model. Specifically, as show in Fig 3, we utilize mathematical calculation formulas as the replaceable parts, with task requirements for direct computation and code programming. For example, subsample 1 "provides the correct answer for 1+1=?" and subsample 2 "programs Python code for 1+1=?".

## 4.2 POSITIONING DECOUPLING DATA

Given an comparative prompt $\{x_k^t\}_{t=1}^2$, we first define the model output $P_{x_k^t, y_t^*}(\widehat{\omega_{X_t}^{l,j}})$ as the probability of the correct answer predicted by a pretrained model:

$$P_{x_k^t, y_t^*}(\widehat{\omega_{X_t}^{l,j}}) = p(y_t^*|x_k^t, \omega_{X_t}^{l,j}[x_k^t] = \widehat{\omega_{X_t}^{l,j}}), \tag{4}$$

where $y_k^*$ denote the correct answer; $\omega^{l,j}$ denotes the $j$-th intermediate neuron in the $l$-th FFN; $\widehat{\omega_{X_t}^{l,j}}$ is a given constant that $\omega_X^{l,j}[x_k^t]$ is assigned to. In order to calculate the attribution score of a neuron $Attr(\omega^{l,j}|x_k^t)$, we gradually change $\omega_{X_t}^{l,j}[x_k^t]$ from 0 its original value $\overline{\omega_{X_t}^{l,j}[x_k^t]}$ calculated by the pretrained model, and meanwhile integrate the gradients:

$$Attr(\omega^{l,j}|x_k^t) = \overline{\omega_{X_t}^{l,j}[x_t]} \int_{\alpha=0}^1 \frac{\partial P_{x_k^t, y_t^*}(\widehat{\omega_{X_t}^{l,j}})}{\partial \omega_{X_t}^{l,j}[x_k^t]} d\alpha \tag{5}$$

where $\frac{\partial P_{x_k^t, y_t^*}(\widehat{\omega_{X_t}^{l,j}})}{\partial \omega_{X_t}^{l,j}[x_k^t]}$ calculates the gradient of the model output with regard to $\omega^{l,j}$. As $\alpha$ changes from 0 to 1, by integrating the gradients, $Attr(\omega^{l,j}|x_k^t)$ accumulates the output probability change caused by the change of $\omega_{X_t}^{l,j}[x_k^t]$. If the neuron has a great influence on the expression of a fact, the gradient will be salient, which in turn has large integration values. Therefore, the attribution score can measure the contribution of the neuron $\omega^{l,j}$ to the factual expressions.

Obtain the coincidence rate $C_r^{t,t^*}$ of the parameter set $P_s$ by comparing the parameters of the sample localization

$$C_r^{t,t^*} = \left\{ \begin{array}{ll} \frac{1}{n \cdot \mathcal{L} \cdot \mathcal{J}} \sum_{k=1}^n (P_s|Attr(\omega^{l,j}|x_k^t)) \cap (P_s|Attr(\omega^{l,j}|x_k^{t^*})) & t \neq t^* \\ \frac{1}{\mathcal{L} \cdot \mathcal{J}} \bigcap_{k=1}^n (P_s|Attr(\omega^{l,j}|x_k^t)) & t = t^* \end{array} \right. \quad t, t^* \in \{1, 2\} \tag{6}$$

where the $P_s$ refers to the located parameters. The coincidence rate $C_r^{1,2}$ of response parameters between subsample 1 and subsample 2 is found to be **15.6%**. Interestingly, we find that the

coincidence rate of response parameters for all subsample1 was **7.3%**, which is **8.6%** for all subsample2. Individual knowledge cannot achieve parameter localization, can the commonality of data be localized?

# 5 IDENTIFYING CAPABILITY NEURONS

In this section, we propose a Commonality Neurons Locating (CNL) method. By utilizing the parameters of localization, we can significantly enhance the corresponding capabilities of the model.

## 5.1 CNL METHOD

To find out the capability neurons under the dataset $\mathcal{D} = \{(x = [x_1, \cdots, x_X], y = [y_1, \cdots, y_Y])\}$, we expand the KN method (Dai et al., 2021; Yao et al., 2024) to compute the contribution of the neuron as:

$$Score(\omega^{l,j}) = \mathbb{E}_{(x,y)\in\mathcal{D}} \left[ \frac{1}{Y}\frac{1}{S} \sum_{m=1}^{Y} \overline{\omega_{Z_m}^{l,j}[z_m]} \sum_{n=0}^{S} \frac{\partial P_{z,y_m}(\frac{n}{S}\overline{\omega_{Z_m}^{l,j}[z_m]})}{\partial \omega_{Z_m}^{l,j}[z_m]} \right], \tag{7}$$

$$z_m = x \oplus y_{0:m-1}$$

where $\oplus$ means a splice of two text. The above equation (7) simulates the expectation of gradient at different outputs of the neuron under the dataset $\mathcal{D}$. In our experiment we let step $S = 19$. To identify the commonality neuron, we take the $Mask$ matrix:

$$Mask_{l,j} = \begin{cases} 1 & \left| Score(\omega^{l,j}) - mean(Score(\omega)) \right| > \sigma \cdot var(Score(\omega)) \\ 0 & else \end{cases} \tag{8}$$

where $mean(\cdot)$ denotes the mean value of all scores and $var(\cdot)$ indicates the variance of the neurons. $\sigma$ is the threshold guiding us to find the commonality neurons. In the absence of any special instructions to follow, we view the neurons with scores outside $\sigma = 6$ as commonality neurons.

## 5.2 EXPERIMENT

Our experiment mainly has the following two findings: 1) We find that a set of data has certain commonalities, which are reflected in the manner of neurons. 2) This commonality crosses over datasets, reflecting the capabilities of the model.

**Datasets.** In the experiment, we utilize three types of datasets that reflect the commonality on the data.

- Math: (1) GSM8K (Cobbe et al., 2021) contains approximately 8,000 elementary math problems with detailed solutions, designed to train mathematical reasoning models; (2) Meta_Math (Yu et al., 2023) focuses on meta-learning for math problems, aims at enhancing the model's adaptive learning and reasoning capabilities. In order to solve this task, we need the model to be mathematically competent, as well as have some multiple choice and language comprehension skills.

- Program: Code25K (Beguš, 2021) contains around 25,000 code snippets, supporting tasks like code generation and completion. In order to solve this task, we need the model to have program capability along with comprehension.

- Language: (1) Emotion (Kosti et al., 2019) with text data labeled with various emotions, suitable for sentiment analysis tasks, including social media posts and comments; (2) Imdb (Tripathi et al., 2020) contains movie reviews and ratings, widely utilized for sentiment analysis and recommendation system research. In order to accomplish this task, we need the model to be capable of sentiment analysis, as well as some multiple choice and language comprehension.

## 5.3 COMMONALITY NEURON EXPERIMENT

**Commonality Neuron Locating.** We utilize 1600 GSM8K, 2400 Emotion, 1200 Code25K, 700 Meta_Math and 800 Imdb data for commonality neuron locating. When locating neurons, we divide the entire dataset into two subsets, $a$ and $b$, and calculate the $overlap$ and $IoU$. The indicator $neuron$ refers to the proportion of localized parameters to MLP parameters.

$$overlap = \frac{\frac{|a \cap b|}{|a|} + \frac{|a \cap b|}{|b|}}{2}, IoU = \frac{|a \cap b|}{|a \cup b|},$$

$$neuron = \frac{\sum_{l=1}^{\mathcal{L}} \sum_{j=1}^{\mathcal{J}} Mask_{l,j}}{\mathcal{L} \cdot \mathcal{J}}$$

(9)

The results obtained are shown in Table 1. We find that both $overlap$ and $IoU$ have high values (like 96.42% and 95.95%), indicating a set of data has certain commonalities, which is reflected by the neurons. At the same time, the low number of $neuron$ indicates a granularity matching of parameters and commonalities.

| Model | ratio | GSM8K | Emotion | Code25K | Meta_Math | Imdb |
|---|---|---|---|---|---|---|
| | $overlap$ | 96.42 | 97.93 | 90.14 | 94.33 | 95.81 |
| Llama2-7B | $IoU$ | 93.08 | 95.95 | 82.05 | 89.15 | 91.96 |
| | $neuron$ | 0.14 | 0.19 | 0.11 | 0.14 | 0.19 |
| | $overlap$ | 94.26 | 94.68 | 91.10 | 95.37 | 98.32 |
| Llama2-13B | $IoU$ | 88.92 | 89.79 | 83.62 | 91.10 | 96.68 |
| | $neuron$ | 0.10 | 0.19 | 0.11 | 0.09 | 0.08 |
| | $overlap$ | 95.66 | 87.62 | 91.25 | 83.62 | 98.27 |
| GPTJ-6B | $IoU$ | 91.63 | 77.96 | 83.89 | 71.77 | 96.60 |
| | $neuron$ | 0.28 | 0.19 | 0.11 | 0.27 | 0.26 |

Table 1: Neurons overlap ratio and the proportion of targeted neurons.

The experimental results above indicate that the neurons we find can reflect a commonality in the data. This commonality is not only confined to a single data, but reflects a shared attribute of a dataset. It is independent of our partitioning method of the dataset. And the Appendix E illustrates the change in localization accuracy with the variation in the amount of data.

## 5.4 CAPABILITY NEURON EXPERIMENT

In the following two experiments, we prove that the commonality neurons are related to the performance of the model and validate the potential of CNL.

**Proving I: Enhance Experiment.** The above experiment demonstrates that we have successfully located neurons with commonality. To verify the effectiveness of the commonality neurons, we conduct enhance experiments. Specifically, we fine-tune on the training set and evaluate on the validation set. Adam (Kingma, 2014) is selected as the optimizer algorithm with lr=1e-5, and the optimized parameters are set as follows:

- Random: We randomly select neurons that are consistent with the number of the located neurons, the proportion of occupying the overall parameters of the neurons is 0.15%.

- W/o located: We mask the located neurons (set their parameter to 0) and fine-tune all other neurons, the proportion of occupying the overall parameters of the neurons is 99.85%.

- Located: We fine-tune the located neurons, the proportion of occupying the overall parameters of the neurons is 0.15%.

| Model | Method | epoch = 1 | | | | epoch = 5 | | | | epoch = 10 | | | |
|---|---|---|---|---|---|---|---|---|---|---|---|---|---|
| | | GSM8K | Emotion | Code25K | Avg. | GSM8K | Emotion | Code25K | Avg. | GSM8K | Emotion | Code25K | Avg. |
| Llama2-7B ($\sigma = 6$) | Random | 0.00 | 14.62 | 52.79 | 22.47 | 0.02 | 14.62 | 52.90 | 22.51 | 5.25 | 14.99 | 53.05 | 24.43 |
| | W/o located | **25.35** | 19.06 | 44.43 | 28.95 | 24.44 | **39.93** | 45.63 | 36.67 | 25.06 | **46.48** | | 40.51 |
| | Located | 24.52 | 23.57 | 54.28 | **34.12** | **24.79** | 32.33 | 55.57 | **37.56** | **25.75** | 44.93 | 55.68 | **42.12** |
| Llama2-7B ($\sigma = 3$) | Random | 0.00 | 14.04 | 52.88 | 22.31 | 24.31 | 22.38 | 53.37 | 33.35 | 23.75 | 26.79 | 53.47 | 34.67 |
| | W/o located | **24.56** | 18.38 | 39.37 | 27.44 | 25.31 | 18.29 | 41.48 | 28.36 | 25.19 | 19.29 | 42.77 | 29.08 |
| | Located | 23.44 | **30.46** | 54.63 | **36.18** | **25.81** | **46.04** | 55.93 | **42.59** | **26.31** | **51.62** | 56.02 | **44.65** |
| GPTJ-6B ($\sigma = 3$) | Random | 0.00 | 5.71 | 50.81 | 18.84 | 25.94 | 23.42 | 50.93 | 33.43 | 25.69 | 28.54 | 51.07 | 35.10 |
| | W/o located | 23.31 | **31.00** | 43.73 | 32.68 | 26.25 | **33.67** | 47.37 | 35.76 | 32.00 | 38.71 | 48.50 | 39.73 |
| | Located | **24.75** | 28.00 | 51.48 | **34.74** | **26.38** | 31.50 | 52.42 | **36.77** | 27.38 | 48.58 | 52.53 | **42.83** |

Table 2: Enhancement of different sets of neurons. The best results are in **bold** and underline means the suboptimal.

As shown in Table 2, the enhancement experiment is validated on two models, llama2 and GPTJ. Compared to random and w/o located, our located has achieved superior performance in multiple datasets. For example, the llama2-7B model achieves **5.17%** improvement in the $Avg$ metric for epoch 1, while the GPTJ-6B model achieves **9.87%** improvement in the Emotion dataset for epoch 10. We can find that we only need to update a small portion of parameters to effectively improve the performance of the current task. We also plot the loss curve for Llama2-7B ($\sigma = 6$) in Appendix F.

**Proving II: Erase Experiment.** To further verify the correlation between commonality neurons and model performance, we conduct erasure experiments. Specifically, we choose different thresholds ($\sigma \in [3, 6, 12]$) to compare the performance changes.

| Model | | Llama2-7B | | | | Llama2-13B | | | |
|---|---|---|---|---|---|---|---|---|---|
| Dataset | | GSM8K | Emotion | Code25K | avg. | GSM8K | Emotion | Code25K | avg. |
| Base accuracy | | 0.00 | 17.63 | 40.80 | 19.48 | 0.38 | 31.96 | 45.95 | 26.10 |
| $\sigma = 3$ | Random | 0.00 (↓ 0.00) | 17.38 (↓ 0.25) | 40.59 (↓ 0.21) | 19.32 (↓ 0.16) | 1.00 (↑ 0.62) | 30.96 (↓ 1.00) | 45.72 (↓ 0.23) | 25.89 (↓ 0.21) |
| | Locate | 0.00 (↓ 0.00) | 0.00 (↓ 17.63) | 25.86 (↓ 14.94) | 8.62 (↓ 10.86) | 0.00 (↓ 0.38) | 0.33 (↓ 31.63) | 21.95 (↓ 24.00) | 7.42 (↓ 18.68) |
| $\sigma = 6$ | Random | 0.00 (↓ 0.00) | 17.50 (↓ 0.13) | 40.67 (↓ 0.13) | 19.39 (↓ 0.09) | 0.31 (↓ 0.07) | 33.75 (↑ 1.79) | 45.81 (↓ 0.14) | 26.62 (↑ 0.52) |
| | Locate | 0.00 (↓ 0.00) | 3.38 (↓ 14.25) | 32.77 (↓ 8.03) | 12.05 (↓ 7.43) | 0.00 (↓ 0.38) | 5.38 (↓ 26.58) | 18.06 (↓ 27.89) | 7.81 (↓ 18.29) |
| $\sigma = 12$ | Random | 0.00 (↓ 0.00) | 17.54 (↓ 0.09) | 40.79 (↓ 0.01) | 19.44 (↓ 0.04) | 0.38 (↓ 0) | 32.04 (↑ 0.08) | 45.80 (↓ 0.15) | 26.07 (↓ 0.03) |
| | Locate | 0.06 (↑ 0.06) | 10.38 (↓ 7.25) | 34.11 (↓ 6.69) | 14.85 (↓ 4.63) | 0.31 (↓ 0.07) | 2.50 (↓ 29.46) | 20.48 (↓ 25.47) | 7.76 (↓ 18.34) |

Table 3: Erase of different sets of neurons. The values represent the comparison between the performance of the current model and the base model after erasing the located neurons.

As shown in the Table 3, compared to random, erasing the neurons we locate will significantly impair the performance of the model. For example, the Llama2-13B model shows a **18.68%** decrease in performance when $\sigma = 3$. This means that our method can locate the capability of the model on a single dataset, which indicates that commonality reflects the capabilities of the model. However, capability be an attribute that can span datasets. Furthermore, we will explore whether the neurons we located can perform across datasets.

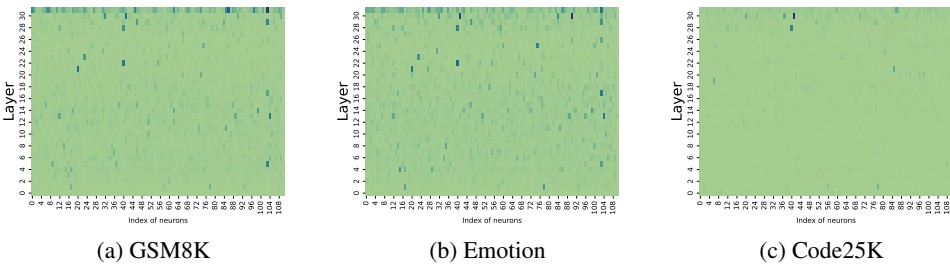

(a) GSM8K     (b) Emotion     (c) Code25K

Fig. 4: Visualisation of commonality neurons. For ease of observation, we choose the neuron with the highest absolute value among 100 neighboring neurons. The horizontal axis shows neuron IDs, the vertical axis shows model layer IDs, and darker squares indicate more prominent located neurons.

**Commonality Across Datasets.** Firstly, we visualize the neurons we locate. From Fig 4, it seems that neurons under different datasets have overlaps, which means that the commonality have cross

dataset characteristics. To further substantiate this viewpoint, we enhance and erase the neurons located on GSM8K, Emotion, and Code25K, and test the performance changes of the model on other datasets.

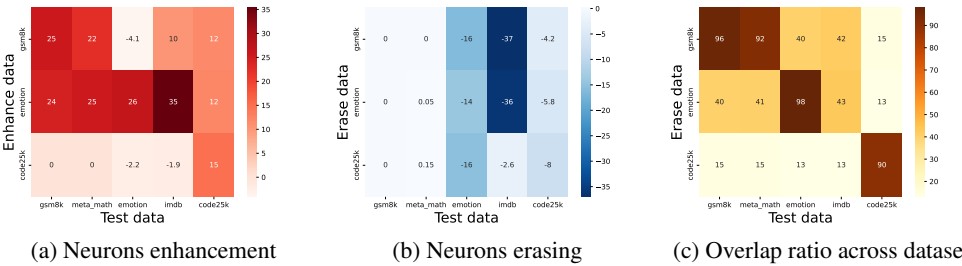

(a) Neurons enhancement      (b) Neurons erasing      (c) Overlap ratio across dataset

Fig. 5: Effects of capacity neurons on other dataset. The vertical axis represents the enhanced or erased dataset, and the horizontal axis represents the tested dataset. The number represents the performance difference between the enhanced or zeroed model and the base model.

The Fig 5 shows that the located neurons have cross-dataset characteristics. Experimental results on additional models are presented in Appendix G. For example, in Fig 5a, enhancing the neurons for GSM8K localization will significantly improve the performance of mathematical related datasets (meta_math) by **22%**, with little impact on other datasets. In the Appendix H, we also calculate the average performance improvement on other datasets not involved in training. The results indicate that, on average, enhancing commonality neurons can bring about more generalized capability improvement. And the detailed results are shown in Appendix I. These results show that the neurons we have located can reflect the cross-data capability of the model. An interesting point is that the GSM8K and Emotion's neurons have a higher overlap rate, which is because there is a greater overlap in capacity required for GSM8K (math, multiple choice, comprehension, etc.) and Emotion (sentiment analysis, multiple choice, comprehension, etc.). Therefore, we can assert that **the located neurons embody the collection of capabilities**. Additionally, to demonstrate the potential of CNL, we conducte a comparison of existing parameters localization methods.

**Comparison of Different Localization Methods.** We construct three unified evaluation indicators, including the consistency of knowledge localization (**Overlap**), the granularity of identified parameters (**Neuron**), and the **I**mpact on model **P**erformance when manipulating identified **P**arameters (**IPP**) for comparison. The calculation formula for this IPP is: Performance (parameters for operation localization) - Optimal-performance (random parameters for multiple sampling). The result from Table 4 proves that previous research (individual knowledge localization) is unreasonable, which also brings new perspectives to subsequent research: focusing on the localization of model capabilities.

| Method | Overlap (↑) | Neuron | IPP (↑) |
|--------|-------------|--------|---------|
| KN | 37.3 | 0.2 | 11.4 |
| ROME | 32.7 | 16.7 | 0.0 |
| KC | 7.2 | 1.6 | 10.5 |
| CNL(Ours) | **96.4** | 0.2 | **20.0** |

Table 4: Comprehensive analysis of different methods. The superior indicators highlighted in **bold**.

## 6 CONCLUSION

In this Paper, we aim to clarify whether existing individual knowledge storage is correct and clarify that capabilities can be localized. Firstly, we demonstrate through fidelity and reliability experiments that the existing knowledge localization methods are unreasonable. In order to further reveal the form of knowledge storage, we found through decoupling experiments that individual knowledge cannot be localized, and the commonality of data has the potential to be localized by parameters. Finally, we propose a commonality neuron localization method that utilizes samples of the same type to obtain commonalities in data and successfully locates commonality neurons. Furthermore, we have demonstrated through cross data experiments that commonality neurons are a collection of capability neurons that possess the capability to enhance performance on other datasets.

ACKNOWLEDGMENTS

This work was supported by the National Key R&D Program of China (No. 2022ZD0116314), Beijing Natural Science Foundation (L243006) and the National Natural Science Foundation of China (No. 62106249).

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

## A APPENDIX / LIMITATIONS

We note a few limitations of the experiments conducted in this paper:

(1) In the experimental section, we utilize the GPT2 model as the base model and did not attempt to evaluate its performance on other models (like Llama2 (Touvron et al., 2023) ).

(2) The maximum parameter size of the tested model is only 7B, and it is possible to attempt capability localization on larger models.

## B APPENDIX / ATTENTION MECHANISM

**Attention mechanism.** Given a sequence of text $x = [x_1, \cdots, x_X]$, the transformer's hidden state $h_i^l$ at the layer $l$ and the token $i$ is calculated:

$$
\begin{aligned}
h_i^l[x] &= h_i^{l-1}[x] + att_i^l[x] + m_i^l[x] \\
att_i^l[x] &= attention^l(h_1^{l-1}[x], \cdots, h_i^{l-1}[x]) \\
m_i^l[x] &= W_{out}^l \sigma(\omega_i^l[x]), \quad \omega_i^l[x] = W_{in}^l \gamma(att_i^l[x])
\end{aligned}
\tag{10}
$$

where $\gamma$ indicate layer norm and $\sigma$ means a non-linear function(Huang et al., 2021b; 2022b). $m_i^l[x]$ could be considered the memory from LLM because feed-forward layers act as key-value mechanism (Dai et al., 2021; Meng et al., 2022b). In order to generate $m_i^l[x]$, the input $k_i^l := \gamma(att_i^l[x])$ activates the corresponding neurons with inner production:

$$
\overline{\omega^{l,j}[x]} = W_{in}^l \cdot k_i^l = [W_{in}^{l,1}, \cdots, W_{in}^{l,N}]^T \cdot k_i^l = [W_{in}^{l,1} \cdot k_i^l, \cdots, W_{in}^{l,N} \cdot k_i^l]^T
\tag{11}
$$

with $\overline{\omega_i^{l,j}[x]} := W_{in}^{l,j} \cdot k_i^l$ standing for the origin output of neuron $W_{in}^{l,j}$. Meanwhile $W_{in}^{l,j}$ is a row vector of $W_{in}$. For brevity, we utilize $W^{l,j}$ to denote $W_{in}^{l,j}$ and $\omega_i^{l,j}[x]$ means the output of $W^{l,j}$ at $i^{th}$ token.

## C APPENDIX/ THE LOCATING METHOD OF PARAMETER LAYERS

In the clean run, they pass a prompt $x = [x_1, ..., x_X]$ into model $\mathcal{F}_\theta$ and collect all hidden activations $\{h_i^l | i \in [1, X], l \in [1, L]\}$, $L$ represents the number of hidden layers in the model.

In the corrupted run, they consider the text before the relationship as the subject, and the text after the relationship as the object. The subject is obfuscated from $\mathcal{F}_\theta$ before the network runs. Concretely, immediately after x is embedded as $[h_1^0, h_2^0, ..., h_T^0]$, we set $h_i^0 = h_i^0 + \delta$ for all indices $i$ that correspond to the subject entity, where $\delta \in \mathcal{N}(0, \sigma^2)$. $\mathcal{F}_\theta$ is then allowed to continue normally, giving us a set of corrupted activations $\{h_{i*}^l | i \in [1, X], l \in [1, L]\}$. Because $\mathcal{F}_\theta$ loses some information about the subject, it will likely return an incorrect answer (Li et al., 2021a; Huang et al., 2022a).

In the corrupted-with-restoration run, they have the $\mathcal{F}_\theta$ run calculations on noise embeddings, except in some tokens $x_{i'}$ and layers $l'$. Afterwards, we hook $\mathcal{F}_\theta$ and forced it to output clean state $h_{i'}^{l'}$. Future calculations can continue without intervention. Afterwards, The capability of a few clean states to restore correct facts afterwards indicates their importance in the calculation graph (Huang et al., 2021a; Weng et al., 2021).

The probability value $P_{l'}$ of restoring the target answer will be utilized as the contribution of this layer $l'$ to common sense knowledge. The larger $P_{l'}$, the greater the probability that individual knowledge is stored in this layer(Li et al., 2021b; Xia et al., 2022).

## D APPENDIX/ THE LOCATING METHOD OF PARAMETER CHAINS

They concentrate on the task of answering factual open-domain questions, where the goal is to predict a target entity $o$ given a subject-relation pair $(s, r)$. Individual knowledge triplet $k = (s, r, o)$ is often presented to the model. To identify the circuit that are critical for predicting the target entity o for a given subject-relation pair $(s, r)$, we ablate each special edge $e_i = (n_x, n_y)$ in the computation graph

$G$. They then measure the impact of ablating the edge (zero ablation in our implementation) on the model's performance using the MatchNLL (Bonvini & Marzani, 2018) loss for the target $o$:

$$S(e_i) = -\log(\frac{G}{e_i(o|(s,r))}) + \log(G(o|(s,r)))) \tag{12}$$

## E APPENDIX/ LOCATING ACCURACY CONVERGENCE

As show in Fig 6, we gradually scale up the amount of data for the localisation neurons and found that the consistency of the localisation converges as the data size increases.

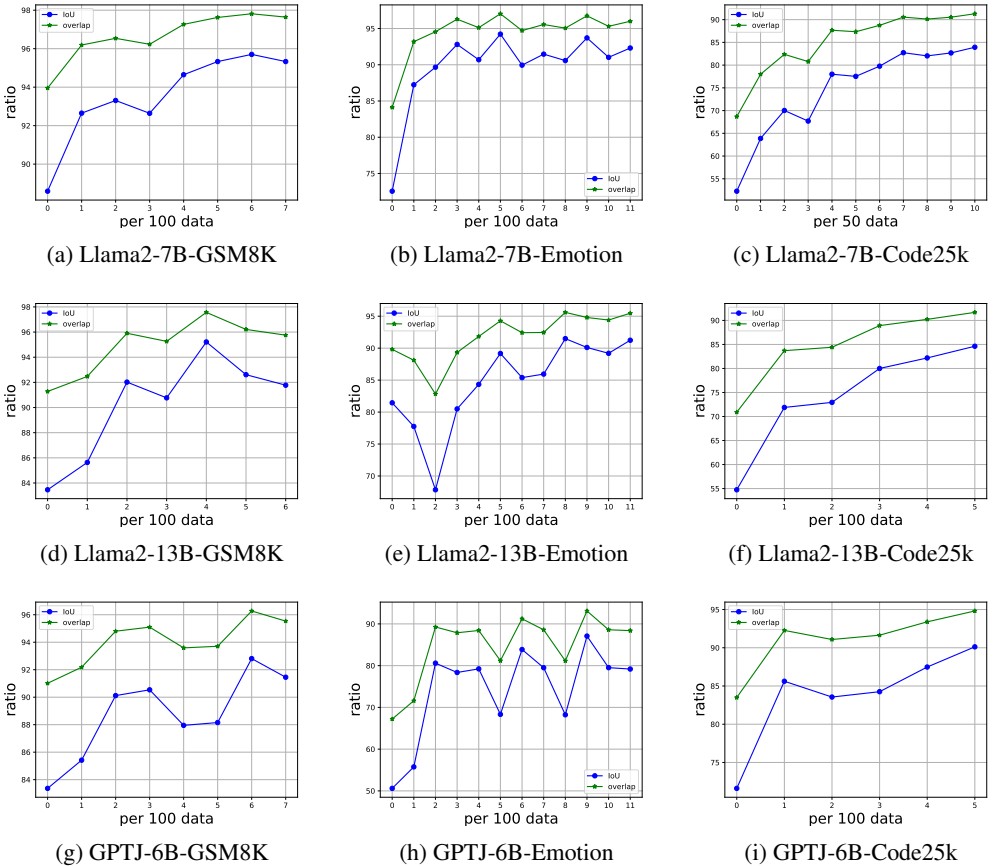

Fig. 6: The relationship between ratio and data. The horizontal axis represents the amount of data utilized for localization, the vertical axis represents locating ratio, which means location accuracy.

The Fig 6 a-c demonstrate that, the location ratio will gradually converge with increasing of data. This suggests that we only need a subset of the dataset to effectively identify the commonality of the entire dataset. Additionally, we concurrently conduct localization experiments under both Llama2-13B and GPTJ-6B. As shown in Fig 6 d-i, for models of different size and architecture, the locational consistency of commonality neurons progressively improves with the increase of data.

## F  APPENDIX/ ENHANCEMENT LOSS WITH DIFFERENT SETS OF NEURONS

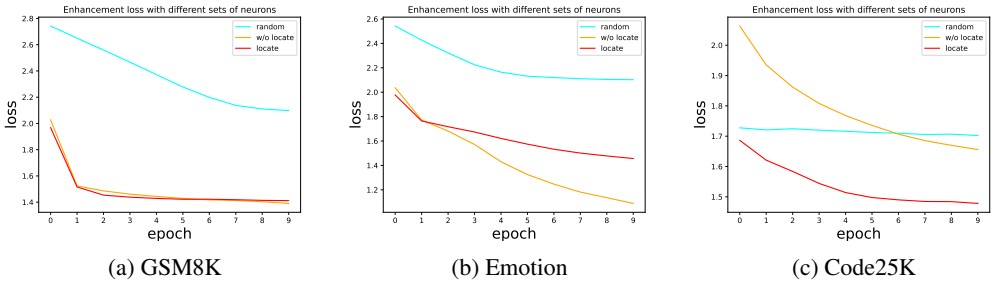

(a) GSM8K          (b) Emotion          (c) Code25K

Fig. 7: Enhancement loss with different sets of neurons.

## G  APPENDIX/ CROSS-DATA RESULT ON OTHER MODEL

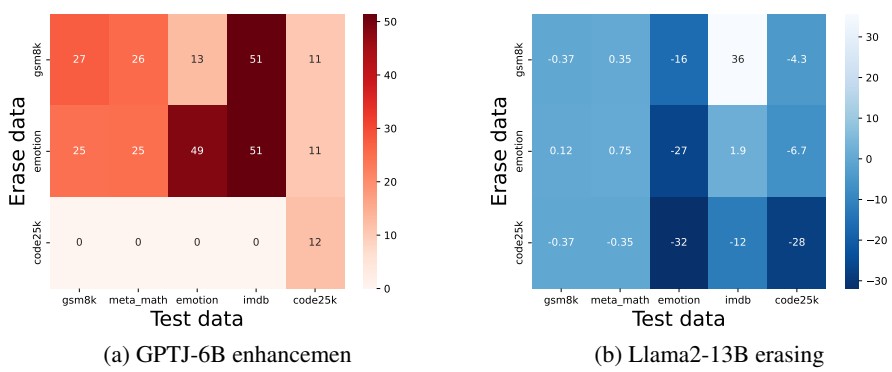

(a) GPTJ-6B enhancemen          (b) Llama2-13B erasing

Fig. 8: Effects of task neurons on other dataset.

## H  APPENDIX/ CHANGES IN MODEL PERFORMANCE ON OTHER DATASETS

We evaluated the image of enhanced neurons on other datasets that are not involved in training. The following represents the performance improvement of the model in other datasets:

| Model | random | w/o located | located |
|---|---|---|---|
| LLama2-7B $\sigma = 6$ | 3.20 | -3.31 | 11.18 |
| LLama2-7B $\sigma = 3$ | 5.20 | -5.58 | 9.43 |
| GPTJ-6B $\sigma = 6$ | 9.00 | 14.73 | 17.66 |

Table 5: The changes in the average performance on other datasets that are not involved in training.

# I  APPENDIX/ CROSS-DATA RESULT ON OTHER METHODS

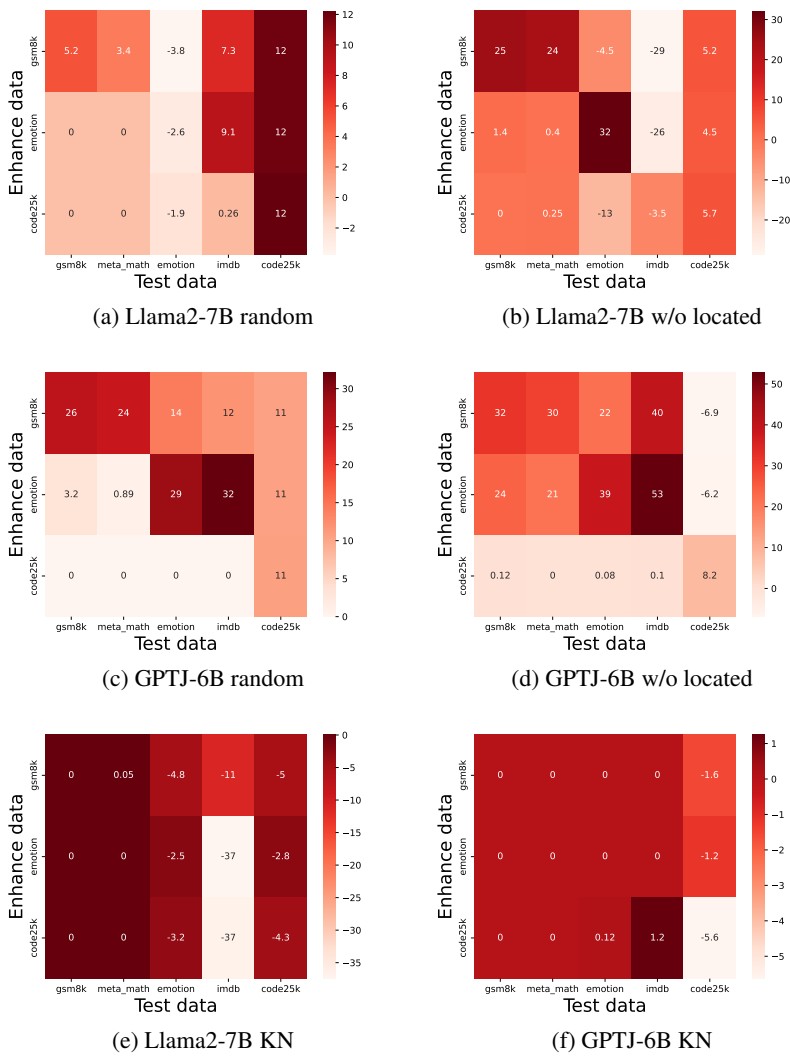

Fig. 9: Effects of task neurons on other methods.

