# OpenReview forum: "Capability Localization: Capabilities Can be Localized rather than Individual Knowledge"
_ICLR.cc/2025/Conference — ICLR 2025 Poster_

### Official Review · Reviewer_wrao · 2024-10-30

**Soundness:** 3
**Presentation:** 3
**Contribution:** 3
**Rating:** 6
**Confidence:** 3

**Summary:**

The authors verify the assumption that knowledge is localized in the model parameter space by evaluating existing methods on a synthetically generated dataset. They falsify this assumption and instead argue that capabilities are localized in model weights and layers instead. Subsequently, the authors propose the Commonality Neuron localization method to find neurons that have commonalities within a variety of datasets. By manipulating these neurons, they causally show that the identified neurons that make up only 0.15% of the parameters significantly affects model performance.

**Strengths:**

-- assessed 3 existing methods to verify knowledge representation in neural networks
-- developed a new method to categorize neurons that are selective to similar information to influence performance
-- The decoupling dataset and experiments are informative.
-- The authors used a wide variety of datasets (math, program, language) to evaluate their proposed CNL method.
-- the authors demonstrated that fine-tuning commonality neuron improved performance within and across datasets, demonstrating their casual role in performance.

**Weaknesses:**

-- for Section 3.1 and 3.2, the authors used GPT4o to generate samples with the same semantic and attempted to identify localized representations in GPTJ. It will be informative to generate samples using GPTJ and verify localization methods on the same model i.e. GPTJ only. If this similarly shows low localization, it would further support the authors' claim.
-- The authors used GPT4o to rewrite 1000 factual statements to populate the factual dataset. How did they verify that the new samples have the same semantic meaning?
-- Why did the authors not evaluate the CNL method on the same dataset i.e. Table 1, that was used to evaluate previous methods?
-- Authors should include the baseline performance of the model without any fine-tuning in Table 2 as in Table 3 for comparison. Is there a claim the authors are making with respect to Table 2?

**Questions:**

-- what is the difference between knowledge and capabilities?
-- Fig 2 is missing its x axis label. I am guessing it is the layer number?

---

> ### Author Response · Authors · 2024-11-19
> **Rebuttal by Authors**
>
> Thank you for the positive recommendations and valuable feedback!
>
> **Q1:** Generate samples utilizing the GPTJ model and validate the localization method on the GPTJ model
>
> **A1:** We generated 1000 test samples utilizing the GPTJ model and validated the localization method on the GPTJ model. The results obtained are as follows. We have constructed three unified evaluation indicators,  including the accuracy of knowledge localization (**Overlap**), the granularity of identified parameters (**Neuron**), and the **I**mpact on model **P**erformance when manipulating identified **P**arameters (**IPP**). The calculation formula for this IPP is: Performance (parameters for operation positioning) - Optimal performance (random parameters for multiple sampling).
>
> |Method|Overlap$\uparrow$|Neuron|IPP$\uparrow$|
> |:-:|:-:|:-:|:-:|
> |GPT4o||||
> |KN [1]|37.3|0.2|11.4|
> |ROME [2]| 32.7|16.7|0|
> |KC [3]|7.2|1.6|10.5|
> |GPTJ||||
> |KN [1]|36.7|0.19|10.1|
> |ROME [2]| 31.8|13.4|0|
> |KC [3]|7.3|1.4|9.7|
> |**CNL(ours)**|**96.4**|**0.2**|**20.0**|
>
> The results showed that our localization method demonstrated similar low localization to previous localization methods, validating our previous conclusion.
>
> **Q2:**  Verify the semantic similarity of 1000 samples
>
> **A2:** To verify the semantic similarity of the new samples, we adopted the following two methods for validation. (1) The Llama38b model was used to evaluate language similarity, and the score obtained was **94.3%**. (2) The semantic similarity formula (Cosine similarity between vectors) was directly used for calculation, and the score obtained was **91.6%**. The experimental results indicate that the new samples have high semantic similarity and meet the requirements of subsequent experiments.
>
> **Q3:** Why hasn't the CNL method been evaluated on a unified dataset
>
> **A3:** The CNL is a method utilized to locate common neurons, specifically, it captures commonalities in batch data with similar types (such as programming-related datasets). However, the previous method focused on locating individual knowledge, which we have proven to be unreasonable in section 3. Importantly, we have constructed a unified evaluation index in A1, which also proves the effectiveness of the CNL.
>
> **Q4:** Is there a claim the authors are making with respect to Table 2
>
> **A4:** Table 2 mainly indicates that the positioning method CNL has high accuracy. Compared with an equal amount of random parameters and 99.85% non-localization parameters, only fine-tuning 0.15% of the parameters can significantly improve the performance of the model, which proves that CNL accurately locates the ability neurons. We have added an additional experiment,  as shown in the table below, utilizing the located neurons for fine-tuning (training data is GSM8K) will cause minimal damage to the performance of other tasks (emotion, imdb, and code25k datasets), and may even benefit other downstream tasks. This provides a new paradigm of "Localization+Fine-tuning" for downstream task applications, greatly increasing practical value in real-world scenarios.
>
> |Model|random|w/o located|located|
> |:-:|:-:|:-:|:-:|
> |LLama2-7B|3.20|-3.31|11.18|
> |GPTJ-6B |9.00|14.73|17.66|
>
> **Q5:** What is the difference between knowledge and capabilities
>
> **A5:** Individual knowledge includes factors such as language, style, semantics, etc., and it is difficult for individual knowledge to be directly mapped to parameters. However, the capability of the model is to learn a large number of sample summaries and inductions, similar to embedding, which is abstract and has the potential to correspond directly to parameters. In addition, our additional experiments have shown that operational capability neurons significantly affect the performance of the current task, which also proves the correctness of studying the localization of capability neurons.
>
>
> (1) Enhance experiment.
>
> llama2-7B
> |Method|GSM8K|Emotion|Code25K|$Avg.$|
> |:-:|:-:|:-:|:-:|:-:|
> |random|23.75|`26.79`|`53.47`|`34.67`|
> |w/o located|`25.19`|19.29|42.77|29.08|
> |located|**26.31**|**51.62**|**56.02**|**44.65**|
>
>  GPTj-6B:
> |Method|GSM8K|Emotion|Code25K|$Avg.$|
> |:-:|:-:|:-:|:-:|:-:|
> |random|25.69|28.54|`51.07`|35.10|
> |w/o located|**32.00**|**38.71**|48.50|`39.73`|
> |located|`27.38`|**48.58**|**52.53**|**42.83**|
>
> (2) Erase experiment.
>
> Llama2-13B
> |Dataset|GSM8K|Emotion|Code25K|$Avg.$|
> |:-:|:-:|:-:|:-:|:-:|
> |Base acc.|0.38|31.96|45.95|26.10|
> |random|0.31 ($\downarrow 0.07$)|33.75 ($\uparrow 1.79$)|45.81 ($\downarrow 0.14$)|26.62 ($\uparrow 0.52$)|
> |locate|0.00 ($\downarrow 0.38$)|5.38 ($\downarrow 26.58$)|18.06 ($\downarrow 27.89$)|7.81 ($\downarrow 18.29$)|
>
> **Q6:** Fig 2 is missing its x axis label. I am guessing it is the layer number?
>
> **A6:** Yes, we have updated the figure in rebuttal revision.
>
> *Ref*
>
> [1] Knowledge neurons in pretrained transformers. EMNLP 2022.
>
> [2] Locating and editing factual associations in gpt. NeurIPS 2022.
>
> [3] Knowledge Circuits in Pretrained Transformers. NeurIPS 2024.

---

> > ### Author Response · Authors · 2024-11-23
> >
> > Dear Reviewer wrao,
> >
> > We appreciate your initial feedback and have addressed your comments in our previous response. If you have any remaining concerns or need further clarification, we welcome your additional input. Thank you for your continued consideration.
> >
> > Sincerely Yours
> >
> > The Authors

---

> ### Comment · Reviewer_wrao · 2024-11-26
> **Maintain score**
>
> Thank you for the additional experiments and clarifications. I am inclined to keep the score as it is. Please include the updates in the revised manuscript. As a future work, you could consider extending this framework to improve the interpretability of what individual neurons are selective for. For instance: Guertler et al. 2024 https://openreview.net/forum?id=01ep65umEr. All the best!

---

> > ### Author Response · Authors · 2024-11-27
> >
> > Dear Reviewer wrao,
> >
> > We appreciate your support and high recognition of our work. We will consider your suggestions and update them in the revised version. If you have any remaining concerns or need further clarification, we welcome your additional input. Thank you for your continued consideration.
> >
> > Sincerely Yours
> >
> > The Authors

---

### Official Review · Reviewer_4N9w · 2024-11-02

**Soundness:** 2
**Presentation:** 1
**Contribution:** 2
**Rating:** 5
**Confidence:** 3

**Summary:**

This work introduces a Commonality Neuron Localization method as an alternative approach to overcome weaknesses of current methodologies in identifying and locating individual factual knowledge in large language models (LLMs) parameters.

After showing that current algorithms that localize individual knowledge do not respect fidelity and reliability requirements, the authors propose a metric to identify capability neurons for a given dataset based on a gradient attribution score of model parameters.

For some datasets, the authors show that Commonality Localization can be related to the performance of the model by showing that enhancing or erasing their role affects the results of the model on the dataset downstream task.

**Strengths:**

The authors propose a change of perspective to address the current limitations of techniques that attempt to localize individual knowledge in LLMs parameters. They propose to focus on a coarser localization task that focuses on common traits specific to a dataset and task.

**Weaknesses:**

The conceptual differences and limitations that incur when replacing individual knowledge location with "commonality neurons" are not sufficiently discussed. These are radically different procedures, with extremely different potential applications. The authors should thoroughly justify why it is relevant to localize phenomena at a much coarser level.

A more thorough discussion of the definition of attribution score would be beneficial, accompanied by a comparison to existing choices for attribution score appeared in the literature.

In the current form, the mapping between commonality neurons and capability is conjectural (see line 525-526). This is not coherent with the current main statement of the work "the located neurons embody the collection of capabilities".

**Questions:**

It is not clear from the results if "Existing technology cannot localize individual model parameters" (line 71) or if "Individual knowledge cannot achieve parameter localization" in general. Clarifying this would be beneficial.

In order to support the claim that "the located neurons embody the collection of capabilities" it should be at least clarified explicitly which capabilities the authors are referring to. In particular, it should be stressed which of the quantitative results in Section 6 support the existence of the specific "common capabilities".

The results of Subsection 5.4 (in particular "Commonality across datasets") should be explained with more care.

The paper could be largely improved in terms of clarity and quality of the presentation:
- extended and more precise Figure captions would help clarify results;
- the role of section 4.2 is not clear given the current form of the paper;
- in formula (8), line 329, the parameter sets $P_s$ should be clearly defined;
- in several circumstances, the authors use terminology incoherent with scientific rigor (e.g. "believes" or "ideologically" referring to existing literature);
- there are several typos and incomplete sentences in the current version.

---

> ### Author Response · Authors · 2024-11-19
> **Rebuttal by Authors**
>
> Thank you for your valuable feedback and for recognizing the novelty of the our method. Below, we address some of the weaknesses raised:
>
> **Q1:** Why it is relevant to localize phenomena at a much coarser level
>
> **A1:** We have constructed three unified evaluation indicators,  including the accuracy of knowledge localization (**Overlap**), the granularity of identified parameters (**Neuron**), and the **I**mpact on model **P**erformance when manipulating identified **P**arameters (**IPP**). The calculation formula for this IPP is: performance (parameters for operation positioning) - optimal performance (random parameters for multiple sampling).
>
> |Method|Overlap$\uparrow$|Neuron|IPP$\uparrow$|
> |:-:|:-:|:-:|:-:|
> |KN [1]|37.3|0.2|11.4|
> |ROME [2]| 32.7|16.7|0|
> |KC [3]|7.2|1.6|10.5|
> |**CNL(ours)**|**96.4**|**0.2**|**20.0**|
>
> From the perspective of consistency (metric **Overlap**), the rewritten individual knowledge is not faithful to the original localization results, and the overlap of the model's capability localization can reach 96.4%.
> From a granularity perspective (metric **Neuron**), the number of parameters occupied by a single knowledge is unreasonable (for example, ROME's Neuron reaches 16.7%).
> From the perspective of operational localization parameters (metric **IPP**), the impact of operating a single knowledge localization neuron on knowledge is not significant, while the impact of capability neurons on capability can reach 20.0%, which also proves that the localization results of a single knowledge are inaccurate.
>
> Individual knowledge (or a prompt), includes factors such as language, style, semantics, etc., and it is difficult for Individual  knowledge to be directly mapped to parameters. However, the capability of the model is to learn a large number of sample summaries and inductions, similar to embedding, which is abstract, so it has the potential to directly correspond to parameters.
> For example, our experiment below shows that enhancing capability neurons improves other tasks that are not involved in training:
>
> |Model|random|w/o located|located|
> |:-:|:-:|:-:|:-:|
> |LLama2-7B $\sigma=3$|5.20|-5.58|9.43|
> |LLama2-7B $\sigma=6$|3.20|-3.31|11.18|
>
> The result have shown that operational capability neurons have a relatively small impact on the performance of the model on other tasks, which also proves the correctness of studying the localization of capability neurons.
>
> **Q2:** Further discussion on attribution scores and comparison with existing methods
>
> **A2:** The previous attribution score (KN) referred to locating individual knowledge, while CNL used integral gradients to obtain the common characteristics of batch knowledge and corresponding parameters. To our knowledge, CNL is the first batch knowledge-based localization method. We listed comparative experiments in **A1**, and the excellent performance proved that previous research (individual knowledge localization) was unreasonable, which also brought new perspectives to subsequent research: focusing on the localization of model capabilities.
>
> **Q3:** It is not clear from the results if "Existing technology cannot localize individual model parameters" or if "Individual knowledge cannot achieve parameter localization"
>
> **A3:** We believe that existing methods cannot accurately locate individual knowledge. However, certain inspirations can be drawn from our experimental results: about 0.15% of the model parameters can reflect a group's capabilities. More fine-grained knowledge should be represented by fewer parameters, while the smallest granularity in current methods is neurons. We do not deny that individual knowledge can be located, but we believe that it should be within the parameters of a smaller granularity.
>
> **Q4:** Inconsistent expression in lines 525-526
>
> **A4:** We have revised the previous statement and added the following enhancement and erasure experiments, which demonstrate a strong correlation between the neurons we locate and their capabilities.
>
>
> (1) Enhance experiment. The best results are in **bold** and `highlighted` means the suboptimal.
>
> llama2-7B
> |Method|GSM8K|Emotion|Code25K|$Avg.$|
> |:-:|:-:|:-:|:-:|:-:|
> |random|23.75|`26.79`|`53.47`|`34.67`|
> |w/o located|`25.19`|19.29|42.77|29.08|
> |located|**26.31**|**51.62**|**56.02**|**44.65**|
>
>  GPTj-6B:
> |Method|GSM8K|Emotion|Code25K|$Avg.$|
> |:-:|:-:|:-:|:-:|:-:|
> |random|25.69|28.54|`51.07`|35.10|
> |w/o located|**32.00**|**38.71**|48.50|`39.73`|
> |located|`27.38`|**48.58**|**52.53**|**42.83**|
>
> (2) Erase experiment.
>
> Llama2-13B
> |Dataset|GSM8K|Emotion|Code25K|$Avg.$|
> |:-:|:-:|:-:|:-:|:-:|
> |Base acc.|0.38|31.96|45.95|26.10|
> |random|0.31 ($\downarrow 0.07$)|33.75 ($\uparrow 1.79$)|45.81 ($\downarrow 0.14$)|26.62 ($\uparrow 0.52$)|
> |locate|0.00 ($\downarrow 0.38$)|5.38 ($\downarrow 26.58$)|18.06 ($\downarrow 27.89$)|7.81 ($\downarrow 18.29$)|

---

> ### Author Response · Authors · 2024-11-19
> **Rebuttal by Authors**
>
> **Q5:** What capability does the author refer to.
>
> **A5:** The capability refers to the ability of  model to solve a specific problem. For example, in the GSM8K dataset, the neurons we locate primarily consists of mathematical and multiple-choice question-solving capability, along with some other minor capabilities such as English understanding.
>
>  Additionally, Fig. 5 indicates that commonality can be inferred by part of the data. This is also a characteristic of capability, that is, we can determine what capabilities are needed to solve the task of the entire data set based on a subset. In Table 2 and Table 3, we demonstrate that the performance of commonality neurons is related to the model, which means it's connected with the model's capability on the current dataset. Fig 6 shows that enhancement and suppression of commonality neurons can also result in increased and decreased performance on similar datasets. This is also consistent with our conclusion: capability is an attribute that is independent of the specific form of data.
>
> **Q6:** About the result of Commonality across datasets.
>
> **A6:** The purpose of conducting cross-dataset experiments is that the model's ability can be cross dataset, and only by achieving significant performance improvements on other datasets, we can clarify that common neurons are the set of corresponding abilities.
>
> We have conducted additional experiments on GPT-J to enhance the persuasiveness of our conclusions. Meanwhile, we provide more detailed explanations of the experiment results in **A4**. Additional across datasets experiment is in APPENDIX H of the rebuttal version.
>
> **Q7:** Other questions about the paper
>
> **A7:** (1) We have increased the clarity of Fig 2 in the article.
>
> (2) The purpose of section 4.2 is to explore potential research directions beyond locating individual knowledge. Interestingly, the experimental results indicate that the localization of capability neurons is valuable,  and provides a foundation for the localization of capability neurons.
>
> (3) The $P_s$ in Formula (8)  is the set of parameters for localization.
>
> (4) We have revised the terminology incoherent with scientific rigor.
>
> We promise that we have fixed the above issues in the rebuttal version.
>
>
> *Ref*
>
> [1] Knowledge neurons in pretrained transformers. EMNLP 2022.
>
> [2] Locating and editing factual associations in gpt. NeurIPS 2022.
>
> [3] Knowledge Circuits in Pretrained Transformers. NeurIPS 2024.

---

> > ### Author Response · Authors · 2024-11-23
> >
> > Dear Reviewer 4N9w,
> >
> > We appreciate your initial feedback and have addressed your comments in our previous response. If you have any remaining concerns or need further clarification, we welcome your additional input. Thank you for your continued consideration.
> >
> > Sincerely Yours
> >
> > The Authors

---

> > > ### Author Response · Authors · 2024-11-29
> > >
> > > Dear Reviewer 4N9w,
> > >
> > > We appreciate your feedback and have addressed your questions. Meanwhile, Reviewers Waro and Skue have expressed strong recognition of our paper and increased their scores. If you have any additional questions or need further clarification, we welcome your supplementary comments. Thank you for your continued consideration. Happy Thanksgiving and best wishes!
> > >
> > > Sincerely Yours
> > >
> > > The Authors

---

> > > > ### Author Response · Authors · 2024-12-02
> > > >
> > > > Dear Reviewer 4N9w,
> > > >
> > > > Thank you for your valuable questions. We have carefully addressed all the points you raised! As we approach the end of the discussion period, please feel free to share any additional questions or concerns you may have. We are happy to provide further clarification and welcome your continued engagement.
> > > >
> > > > Sincerely,
> > > >
> > > > The Authors

---

> > > > > ### Comment · Reviewer_4N9w · 2024-12-02
> > > > >
> > > > > Dear Authors,
> > > > > Thank you for addressing the points discussed in my review.
> > > > > I am convinced that some of the directions pursued in the manuscript are interesting, and some of the experiments performed sufficiently support part of the claims stated in the work.
> > > > > Despite this, I still believe the clarity of the writing is still not sufficient, that some definitions (eg "capability") have not been sufficiently described, and that the experiments should be better organized.
> > > > > For these reasons, I will raise my score to 5.

---

> > > > > > ### Author Response · Authors · 2024-12-02
> > > > > >
> > > > > > Dear Reviewer 4N9w,
> > > > > >
> > > > > > Thank you for your response and score increase. In response to your question, we have made the necessary corrections in the rebuttal version. Meanwhile, we promise to update in the revised version. All the best！
> > > > > >
> > > > > > Sincerely,
> > > > > >
> > > > > > The Authors

---

### Official Review · Reviewer_Skue · 2024-11-04

**Soundness:** 2
**Presentation:** 2
**Contribution:** 3
**Rating:** 6
**Confidence:** 2

**Summary:**

This  paper explores the idea of localising “capabilities” within LLMs, rather than “individual knowledge”. The argument is that (existing methods for localizing) individual knowledge is unreliable. For instance, the authors show that current methods like parameter layers and chins are perfectly faithful to individual knowledge. For example, rewriting semantically similar prompts did not lead to consistent localisation results, and manipulating identified parameters did not reliably produce expected changes in model behaviour.

Instead the paper forwards a new framework for localizing “commonality neurons”, the neurons that represent shared attributes or capabilities across datasets. They introduce the **Commonality Neuron Locating (CNL) method** which identifies neurons that consistently activate for similar tasks or data types. The paper concludes that capabilities can be localised within LLMs, potentially leading to new ways of understanding and manipulating LLM functionality

**Strengths:**

1. The paper is clear and well motivated. It clearly showcases the existing methods for understanding LLMs and their limitations.
2. The paper also addresses an important question in LLM research. How LLM parameters relate to their performance is topical. The paper challenges the existing methods providing compelling evidence.
3. The paper also introduces a way of plugging those gaps. The CNL approach is a clear structured approach and inspired by well-grounded methods like integrated gradients from CV.

**Weaknesses:**

1. The usage of arbitrary thresholds (magic numbers): The study uses a threshold of σ = 6 for identifying capability neurons without sufficient justification or exploration of alternative values.
2. The decoupling experiment  to investigate the localisation of commonalities, primarily focuses on mathematical tasks. This narrow scope limits the generalisability of the findings regarding the potential for localising data commonalities. Expanding the decoupling experiment to include tasks from other domains, such as emotion recognition or language understanding, is crucial for supporting broader claims about the relationship between data commonalities and parameter localisation.
3. The study shows that enhancing localised neurons leads to performance improvements, but the statistical significance of these improvements is not adequately assessed. Merely stating that the located neurons are "most sensitive" to performance improvement without providing statistical evidence leaves the strength of this claim open to question.
4. The biggest weakness though is the glaring lack of common evaluation metrics. The study evaluates different knowledge localisation methods (ROME, KN, KC, and CNL) using distinct metrics tailored to each method's specific assumptions. Establishing a shared metric that can be applied consistently across all methods would enable a more objective and informative evaluation. Such a metric could consider factors like the accuracy of knowledge localisation, the granularity of identified parameters, and the impact on model performance when manipulating identified parameters.

**Questions:**

1. The authors should provide justification for the specific hyperparameters chosen and clarify the methodology in more detail. For instance, in Section 5.1, line 352, why is S=19? Was this value selected through a hyperparameter sweep? If so, please include these details in the appendix. Similarly, in Section 5.1, line 358, why is the attribution score threshold set at σ=6? Offering detailed steps for identifying commonality neurons is essential to ensure that the results are not cherry-picked. This transparency will also help reproduce the findings independently.
2. For the random-neuron conditions in the “enhance” and “erase” experiments ([Table 2] and [Table 3]), did you evaluate across multiple random samples to ensure statistical robustness? If so, please indicate the number of samples tested; without this, the reliability of comparisons with targeted neurons remains unclear.
3. We need some more control conditions to convincingly demonstrate the impact of commonality neurons. For instance, a comparison with multiple sets of randomly selected neurons, averaged over several trials, would help determine if performance changes are indeed specific to commonality neurons rather than general effects from a neuron subset. Testing different thresholds for neuron activation and varying the size of neuron groups would also clarify whether the observed effects are specific to the identified commonality neurons. Such controls would strengthen the claim that these neurons contribute uniquely to model capabilities rather than reflecting random network fluctuations.
4. By far the biggest thing missing from the paper is a common metric for comparing different methods. It is crucial to benchmark CNL against established knowledge and capability localization methods, such as distributed parameters, parameter layers, and parameter chains. This comparison should involve measuring CNL’s accuracy, consistency, and computational efficiency relative to these approaches in reproducible way such that improvements can be tracked carefully.

5a. Another major concern is the scalability of CNL. There are at least 3 issues with model scalability and the authors need to discuss (or even better perform new tests) the scalability strategies of CNL.  A) As new models grow larger (e.g., from billions to hundreds of billions of parameters), the CNL method’s neuron-level analysis could become computationally prohibitive.

5B. Larger models might need different threshold and hyperparameter sweeps to maintain CNL’s accuracy which demand additional tuning efforts.

5C. CNL relies on extensive data from various domains to identify commonality neurons. As model sizes increase, the amount of data needed for localization will also grow.

6. The authors should clarify how capability localization could enhance model interpretability in practical applications. It remains unclear whether the CNL approach addresses the theoretical question of how information is localized within the model or if it offers actionable changes for real-world LLM applications. Specifically, the authors should explain how identifying capability neurons could be leveraged to improve model performance, robustness, or transparency in practical scenarios. Without clear guidance the potential of using CNL in real-world settings is ambiguous.
7. While the paper includes extensive mathematical notation and text, the figures are under-explained and hard to follow. For instance, it is unclear what Figure 1 is intended to convey, the purpose of the red boxes in Figure 3, or the significance of the colors in Figure 4. The authors should enhance these figure descriptions to make the visuals more accessible and informative for readers, clearly explaining all elements, symbols, and color schemes. This will improve readability and ensure that the figures effectively support the paper’s findings.

---

> ### Author Response · Authors · 2024-11-19
> **Rebuttal by Authors**
>
> Thank you for recognizing the importance, effort in method, and applications of our work. We outline our response to the main concerns:
>
> **Q1:** Unified evaluation indicators and results
>
> **A1:**  Thanks again for your advice! We have constructed three unified evaluation indicators,  including the accuracy of knowledge localization (**Overlap**), the granularity of identified parameters (**Neuron**), and the **I**mpact on model **P**erformance when manipulating identified **P**arameters (**IPP**). The calculation formula for this IPP is: performance (parameters for operation positioning) - optimal performance (random parameters for multiple sampling).
>
> |Method|Overlap$\uparrow$|Neuron|IPP$\uparrow$|
> |:-:|:-:|:-:|:-:|
> |KN [1]|37.3|0.2|11.4|
> |ROME [2]| 32.7|16.7|0|
> |KC [3]|7.2|1.6|10.5|
> |**CNL(ours)**|**96.4**|**0.2**|**20.0**|
>
> The excellent performance proves that previous research (individual knowledge localization) is unreasonable, which also brings new perspectives to subsequent research: focusing on the localization of model capabilities.
>
>
> **Q2:** Choice of σ and S
>
> **A2:** In statistics, σ=3 is commonly used to detect outliers [1]. The larger the σ, the more stringent the requirements for locating neurons. In order to better validate the effectiveness of our method, we have chosen a more stringent value of σ=6: we generally consider data with σ>6 as outliers in statistics. However, in response to your suggestion, we have also supplemented experimental results under σ = 3. In addition, S=19 is a setting we adopt from the previous method KN [2]. Here we have listed experiments on fine-tuning and localization accuracy with σ=3, and the more stringent condition of σ=6 can achieve considerable experimental results.
>
>  (1) Fine-tuning & $\sigma = 3$
> |Method|GSM8K|Emotion|Code25K|$Avg.$|
> |:-:|:-:|:-:|:-:|:-:|
> |random|23.75|`26.79`|`53.47`|`34.67`|
> |w/o located|`25.19`|19.29|42.77|29.08|
> |located|**26.31**|**51.62**|**56.02**|**44.65**|
>
> (2) Localization & $\sigma = 3$
> |ratio|GSM8K|Emotion|Code25K|Meta_Math|Imdb|
> |:-:|:-:|:-:|:-:|:-:|:-:|
> |overlap|97.21|97.35|89.88|93.47|98.76|
> |IoU|94.49|94.84|81.61|87.56|97.55|
> |neuron|0.46|0.62|0.30|0.44|0.36|
>
> We have provided experimental results with σ=12 at global rebuttal.
>
> **Q3:** Extending decoupling experiments to tasks in other fields
>
> **A3:** To further demonstrate the effectiveness of decoupling experiments, we constructed decoupling data based on the emotion recognition dataset SemEval [3]. For example, the main part is "William did not complete his homework," and the replaceable parts are "Is this sad or joyful?" and "What should he do next." The experimental results are as follows
>
> |Overlap rate|Calculation & programming|Emotion recognition & reasoning|
> |:-:|:-:|:-:|
> |Sub1 & Sub2|15.6|14.3|
> |Sub1 list|7.3|9.7|
> |Sub2 list|8.6|8.9|
>
> The results indicate that the conclusions in the paper are still met on the emotion recognition dataset, and we will add these results in the rebuttal version.
>
> **Q4:** Statistical evaluation of enhancing local neurons to improve performance
>
> **A4:** We extended the experiment in Table 2 to the GPTJ model, and the results showed that enhancing local neurons on different model architectures can still improve performance. The term 'most sensitive' here refers to the performance improvement that can be achieved by fine-tuning only 0.15% of the located parameters compared to random parameters.
>
> |Method|GSM8K|Emotion|Code25K|$Avg.$|
> |:-:|:-:|:-:|:-:|:-:|
> |random|25.69|28.54|`51.07`|35.10|
> |w/o located|**32.00**|**38.71**|48.50|`39.73`|
> |located|`27.38`|**48.58**|**52.53**|**42.83**|
>
> The best results are in **bold** and `highlighted` means the suboptimal.
>
> **Q5:** About the random-neuron conditions in the “enhance” and “erase” experiments
>
> **A5:** Your suggestion was very helpful, and we have adopted it. We averaged the results through three random experiments and have updated the experimental results in Table 2 in rebuttal revision. The results once again confirmed the previous conclusion. For “erase” experiments, we also supplemented the results of Llama2-13B:
>
> |Dataset|GSM8K|Emotion|Code25K|$Avg.$|
> |:-:|:-:|:-:|:-:|:-:|
> |Base acc.|0.38|31.96|45.95|26.10|
> |random|0.31 ($\downarrow 0.07$)|33.75 ($\uparrow 1.79$)|45.81 ($\downarrow 0.14$)|26.62 ($\uparrow 0.52$)|
> |locate|0.00 ($\downarrow 0.38$)|5.38 ($\downarrow 26.58$)|18.06 ($\downarrow 27.89$)|7.81 ($\downarrow 18.29$)|
>
> Erasing localized neurons can lead to more significant performance degradation.
>
> **Q6:** Regarding providing more control conditions to demonstrate the impact of common neurons
>
> **A6:** We have updated more comprehensive experimental results in Tables 2 and 3, sampling and averaging three groups of random neurons in rebuttal revision. The experimental results demonstrate the significant impact of localization parameters on the model's capability.

---

> ### Author Response · Authors · 2024-11-19
> **Rebuttal by Authors**
>
> **Q7:** The Scalability of CNL Method
>
> **A7:** Firstly, in terms of computational complexity, as the parameter size increases, the computational resources required for CNL inevitably increase. Our experiment was conducted on models such as GPTJ, LLAMA2 7B, and LLAMA2 13b, requiring only 4 * NVIDIA A100 (40 GB) of computing resources to complete. Compared to previous methods of fine-tuning models with full parameters, the computational resources and time consumed by CNL are negligible.
>
> Secondly, regarding the adjustment of hyperparameters, as shown in Eq 9 and 10, our hyperparameters mainly include threshold σ and S. The threshold σ is based on the standard deviation theory and does not require manual adjustment. Experimental results show that as the parameter size increases, the number of neurons located also increases, and the proportion is similar.
>
> (1)  Llama2-7B
>
> |ratio|GSM8K|Emotion|Code25K|Meta_Math|Imdb|
> |:-:|:-:|:-:|:-:|:-:|:-:|
> |neuron|0.14|0.19|0.11|0.14|0.19|
> |number|493|669|387|494|671|
>
> (2)  Llama2-13B
>
> |ratio|GSM8K|Emotion|Code25K|Meta_Math|Imdb|
> |:-:|:-:|:-:|:-:|:-:|:-:|
> |neuron|0.10|0.19|0.11|0.09|0.08|
> |number|552|1050|608|497|442|
>
> Finally, regarding the relationship between data volume and parameter size, as the parameter size changes, we also listed the required data volume for different models.
>
> (1) GPTj-6B
>
> GSM8K:
> |Num_data|100|200|300|400|500|600|700|800|
> |:-:|:-:|:-:|:-:|:-:|:-:|:-:|:-:|:-:|
> |overlap|76.57|92.60|88.24|95.74|95.54|93.63|93.91|95.66|
> |IoU|62.01|86.17|78.87|91.81|91.41|87.98|88.47|91.63|
>
> Emotion:
> |Num_data|100|200|300|400|500|600|700|800|900|1000|1100|1200|
> |:-:|:-:|:-:|:-:|:-:|:-:|:-:|:-:|:-:|:-:|:-:|:-:|:-:|
> |overlap|45.67|79.04|83.85|65.92|77.81|85.85|80.99|77.68|87.82|84.28|91.73|87.62|
> |IoU|29.46|65.26|72.18|49.11|63.64|75.20|68.04|63.43|78.26|72.82|84.70|77.96|
>
>
>
> (2)  Llama2-13B
>
> GSM8K:
> |Num_data|100|200|300|400|500|600|
> |:-:|:-:|:-:|:-:|:-:|:-:|:-:|
> |overlap|85.78|90.45|94.24|91.47|94.07|94.26|
> |IoU|73.97|81.30|89.03|83.76|88.37|88.92|
>
>
> Emotion:
> |Num_data|100|200|300|400|500|600|700|800|900|1000|1100|1200|
> |:-:|:-:|:-:|:-:|:-:|:-:|:-:|:-:|:-:|:-:|:-:|:-:|:-:|
> |overlap|89.51|93.60|87.56|83.94|95.56|95.37|94.21|95.72|91.76|95.75|94.01|94.68|
> |IoU|80.51|87.94|76.35|69.30|91.48|91.11|88.97|91.66|84.22|91.84|88.63|89.79|
>
>
> The experimental results indicate that there is no significant difference in the required amount of data at different parameter scales.
>
> **Q8:** How capability localization could enhance model interpretability in practical applications.
>
> **A8:** Firstly, we conducted a new experiment to demonstrate that CNL improves the interpretability and applicability of the model in practical scenarios. The three indicators refer to fine-tuning random parameters, w/o located parameters, and located parameters, and evaluating the model after fine-tuning located parameters on other tasks. The results indicate that fine-tuning the located parameters can not only improve the performance of the model in current tasks but also contribute to other tasks, which proves that CNL can improve the performance and robustness of the model.
>
> |Model|random|w/o located|located|
> |:-:|:-:|:-:|:-:|
> |LLama2-7B $\sigma=3$|5.20|-5.58|9.43|
> |LLama2-7B $\sigma=6$|3.20|-3.31|11.18|
> |GPTJ-6B $\sigma=6$|9.00|14.73|17.66|
>
>
> At the same time, updating only the located parameters is not only more cost-effective, but also demonstrates the accuracy of capability positioning without harming other tasks. By establishing the correspondence between internal parameters and the different capabilities of the model, the transparency of the model has been improved. We promise to provide parameter correspondences for different models, which will help with interpretability research of the community in practical applications.
>
> To our knowledge, CNL is the first method to implement capability localization and has demonstrated that capability can achieve parameter localization rather than individual knowledge, which will contribute to subsequent interpretability research.
>
> **Q9:** Further explanation on the chart
>
> **A9:** Figure 1 mainly visualizes the reliability and fidelity experiments, and conducts experimental analysis on the previous three forms of knowledge storage. The red box in Figure 3 indicates that there are differences in the parameters used for localization between subsample1 and subsample2. Figure 4 visualizes the localization parameters within the model, with darker-colored areas referring to the localized parameters. We have added this explanation in the rebuttal version.
>
> *Ref*
>
> [1] Linear and Nonlinear Models.Fixed effects, random effects, and mixed models. GALE 2006.
>
> [2] Knowledge neurons in pretrained transformers. EMNLP 2022.
>
> [3] SemEval-2016 Task 5: Aspect Based Sentiment Analysis. Hal.science 2016

---

> > ### Author Response · Authors · 2024-11-23
> >
> > Dear Reviewer Skue,
> >
> > We appreciate your initial feedback and have addressed your comments in our previous response. If you have any remaining concerns or need further clarification, we welcome your additional input. Thank you for your continued consideration.
> >
> > Sincerely Yours
> >
> > The Authors

---

> ### Comment · Reviewer_Skue · 2024-11-27
>
> The reviewers have addressed a number of my concerns. I am happy to increase by score to 6. I do request the authors to spend more time describing the figures and adding figure legends. The paper is quite dense and hard to follow in its current form.

---

> > ### Author Response · Authors · 2024-11-27
> >
> > Dear Reviewer Skue,
> >
> > We appreciate your recognition of our paper and the increase in scores. We have submitted the latest rebuttal version, which updates the detailed description of the figures and legends. Additionally, we promise to update this aspect in the revised version. If you have any remaining concerns or need further clarification, we welcome your additional input. Thank you for your continued consideration.
> >
> > Sincerely Yours
> >
> > The Authors

---

### Official Review · Reviewer_dq76 · 2024-11-04

**Soundness:** 2
**Presentation:** 1
**Contribution:** 3
**Rating:** 6
**Confidence:** 2

**Summary:**

The manuscript investigated the storage of individual knowledge in model parameters, and proposed a Commonality Neuron Localization method that locates commonality neurons.

**Strengths:**

1. The manuscript conducts experiments of previous knowledge localization methods and demonstrates those methods lack reliability and fidelity. The finding is insightful to the community.
2. The manuscript proposed a method that can effectively identify those neurons that embody the model capability, which help the understanding of large-scale neural networks.

**Weaknesses:**

1. It is unclear to me what the author means by "commonality". And what is its relation to knowledge localization?
2. Figure 2 is blurry.
3. In line 182, $e_i$ seems to be undefined.
4. Missing reference in line 216.

**Questions:**

1. line 344, "we expand the KN method", what is the key improvement of CNL compared to KN?
2. What is the conclusion of Table 2?
3. The method successfully identifies those commonality neurons that embody the model capability. However, why is finding these neurons useful? For example, do we have any insights from analyzing the characteristics of the commonality neurons?

---

> ### Author Response · Authors · 2024-11-19
> **Rebuttal by Authors**
>
> Thank you for your constructive feedbacks on the paper! We add detailed explanations for the questions asked in the review.
>
> **Q1:** What does "commonality" refer to, and what is its relation to knowledge localization?
>
> **A1:** In section 5.4, we point out that commonality is the set of capabilities. For example, previous knowledge localization utilized individual knowledge called "one plus one equals two", and found the parameters in the model corresponding to that knowledge. However, for the model, this knowledge is not only its computational capability but also its English, style and semantic capability. Therefore, in the article, we refer to the set of model capabilities as commonalities, rather than just simple computational capabilities, which is more rigorous.
>
> **Q2:** Figure 2 is blurry.
>
> **A2:** We have updated this figure in the rebuttal version.
>
> **Q3:** $e_i$ seems to be undefined.
>
> **A3:**  The definition of $e_i$ is in  line 682 on APPENDIX D. The $e_i$ refers to calculating the edges in graph G. We add an explanation for $e_i$ in rebuttal version.
>
> **Q4:** What is the key improvement of CNL compared to KN method?
>
> **A4:** The key improvement is that CNL has acquired the common feature of batch knowledge, rather than individual knowledge. Specifically, KN refers to the localization of individual knowledge, while CNL utilizes integral gradients to obtain the common characteristics of batch knowledge and corresponding parameters. To our knowledge, CNL is the first batch knowledge-based localization method. We have constructed three unified evaluation indicators,  including the accuracy of knowledge localization (**Overlap**), the granularity of identified parameters (**Neuron**), and the **I**mpact on model **P**erformance when manipulating identified **P**arameters (**IPP**). The calculation formula for this IPP is: performance (parameters for operation positioning) - optimal performance (random parameters for multiple sampling).
>
> |Method|Overlap$\uparrow$|Neuron|IPP$\uparrow$|
> |:-:|:-:|:-:|:-:|
> |KN [1]|37.3|0.2|11.4|
> |ROME [2]| 32.7|16.7|0|
> |KC [3]|7.2|1.6|10.5|
> |**CNL(ours)**|**96.4**|**0.2**|**20.0**|
>
> The excellent performance proves that previous research (individual knowledge localization) is unreasonable, which also brings new perspectives to subsequent research: focusing on the localization of model capabilities.
>
> **Q5:** What is the conclusion of Table 2?
>
> **A5:** Table 2 shows that compared to fine-tuning an equal number of random or non-located parameters, fine-tuning the located parameters can significantly improve performance. The experimental results indicate that: (1) The localization method of CNL is effective, as it identifies the parameters of the common characteristics of batch knowledge within the model. (2) The significant impact of a small number of parameters after localization on the performance, proves that commonality (the set of model capabilities) can achieve parameter localization, which has profound significance. We will add this explanation in the revised version.
>
> To demonstrate the generalization of the results, we conducted the same experiment on the GPTJ model, and the results also confirmed our above conclusion.  (For convenience, we only list the results under the last epoch here, and specific experimental results are in our updated Table2):
>
> |Method|GSM8K|Emotion|Code25K|$Avg.$|
> |:-:|:-:|:-:|:-:|:-:|
> |random|25.69|28.54|`51.07`|35.10|
> |w/o located|**32.00**|**38.71**|48.50|`39.73`|
> |located|`27.38`|**48.58**|**52.53**|**42.83**|
>
> The best results are in **bold** and `highlighted` means the suboptimal.
>
> **Q6:** What is the function of the common neurons found?
>
> **A6:** Firstly, by leveraging the superior performance of common neurons, we have broken the traditional mindset of individual knowledge localization and provided the correct research direction: common neuron localization.
>
> Secondly, the cross dataset experiment in Figure 6 demonstrates that we only need to fine-tuning 0.15% parameters to significantly improve or suppress the model's capability in a certain field. This provides a low-cost and efficient solution for applying the model to downstream tasks.
>
> Finally, as shown in the table below, utilizing the located neurons for fine-tuning (training data is GSM8K) will cause minimal damage to the performance of other tasks (emotion, imdb, and code25k datasets), and may even benefit downstream tasks. This provides a new paradigm of "localization+fine-tuning" for downstream task applications, greatly increasing its practical value in real-world scenarios.
>
> |Model|random|w/o located|located|
> |:-:|:-:|:-:|:-:|
> |LLama2-7B $\sigma=3$|5.20|-5.58|9.43|
> |LLama2-7B $\sigma=6$|3.20|-3.31|11.18|
> |GPTJ-6B $\sigma=6$|9.00|14.73|17.66|
>
> *Ref*
>
> [1] Knowledge neurons in pretrained transformers. EMNLP 2022.
>
> [2] Locating and editing factual associations in gpt. NeurIPS 2022.
>
> [3] Knowledge Circuits in Pretrained Transformers. NeurIPS 2024.

---

> > ### Author Response · Authors · 2024-11-23
> >
> > Dear Reviewer dq76,
> >
> > We appreciate your initial feedback and have addressed your comments in our previous response. If you have any remaining concerns or need further clarification, we welcome your additional input. Thank you for your continued consideration.
> >
> > Sincerely Yours
> >
> > The Authors

---

> > > ### Comment · Reviewer_dq76 · 2024-11-23
> > > **Response by the Reviewer**
> > >
> > > Thank authors for clarifications. I still have some questions regarding your replies
> > >
> > > Q1: It is still unclear to me what kind of "capabilities" those neurons are responsible for. And why the pair datasets you proposed can disentangle them. Could you elaborate more about this?
> > >
> > > Q5: Are there any baselines you could consider for table2? e.g. the methods you have talked about in the previous part of your manuscript.
> > >
> > > Q6: In Fig 6 (a), the last row shows zero or negative correlation (e.g. -1.9, -2.2) between the enhanced datasets and test datasets. Does it conflict with your claim?

---

> ### Author Response · Authors · 2024-11-24
>
> Thank you for your prompt response. Below, I will further elaborate on the question you have raised, highlighting our novel contributions and advantages.
>
> **Reply to Q1:** The capability of neurons responsible for localization using different corpora is different. For example, in Chapter 5.2, our CNL localization method utilizes the GSM8K dataset (math related) to locate neurons. The capability of the neurons we locate is mainly computational, followed by secondary abilities such as English and style, depending on which capabilities of the models are mainly called upon by GSM8K. Our experiment in Figure 6.a further confirms this conclusion, and after enhancing the localization of neurons, the performance of the model on the Meta_math dataset (math related) is significantly improved compared to other unrelated datasets. This indicates that the neurons we locate are primarily responsible for computing capability.
>
> As mentioned above, our located capability is related to the corpus. Specifically, the located capability is mainly the capability of the model to call upon these corpora when encountering them. As shown in Figure 3, based on the method of controlling variables, we control the capabilities called by the model by setting the main parts and replaceable parts, which achieves the decoupling of different capabilities. However, we have demonstrated in Section 3 that utilizing individual sample for localization (previous methods) is not feasible because the ability of individual sample is not significant. Our CNL has achieved accurate localization of computing capability utilizing batch samples (such as GSM8K). Strictly, the neurons we locate inevitably contain some secondary abilities (such as English, style, etc.) because GSM8K is an English corpus, which is also emphasized in line 527. What we locate is a set of abilities, mainly computing capability ( for the GSM8K dataset).
>
> **Reply to Q5:** Thank you for your suggestion, we have adopted it. It should be noted that in Section 2, we introduced the current mainstream methods, namely Distributed Parameters (KN), Parameter Layers (ROME) and Parameter Chains (KC). The required corpus for ROME and KC is strict (in the form of triplets (s, r, o)), and they require the position of the last token of the subject, which does not apply to free-text widely present in real-world scenarios (such as the GSM8K dataset in Table 2). We provide the results of KN as a baseline, as shown below
>
> (1) Llama2-7B
> |Method|GSM8K|Emotion|Code25K|$Avg.$|
> |:-:|:-:|:-:|:-:|:-:|
> |KN|0.00|14.88|36.45|17.11|
> |random| 0.00|14.62 |`52.79` |22.47 |
> |w/o located| **25.35**|`19.06` |44.43 |`28.95` |
> |**located(ours)**|`24.52` |**23.57** |**54.28** |**34.12** |
>
> (2) GPTJ-6B
> |Method|GSM8K|Emotion|Code25K|$Avg.$|
> |:-:|:-:|:-:|:-:|:-:|
> |KN| 0.00| 0.00 |34.65 |11.55 |
> |random|  0.00| 5.71 | `50.81` |18.84 |
> |w/o located| `23.31` |**31.00** |43.73 | `32.68`|
> |**located(ours)**| **24.75** |`28.00` |**51.48** |**34.74** |
>
> The experimental results show that our method is significantly superior to KN. In addition, we have provided unified evaluation metrics in **A4** compared to previous methods, which demonstrates the superiority of our approach.
>
> **Reply to Q6:**
> Thank you for your keen observation. This is not conflicting. In **A6**, we mentioned `'may even benefit downstream tasks'`, the conclusion drawn from evaluating multiple downstream tasks. For example, the value `'9.43'` in **A6** is the average performance growth of other datasets after training separately on GSM8K, Emotion and Code25K. It cannot be denied that there may be performance degradation on a small number of tasks, but the performance damage is minimal. Meanwhile, as shown in the results of **A6**, our method can minimize the damage to other tasks compared to w/o located (`-5.58%` and `-3.31%`), and the experimental effect is significant. We have added specific comparison results in Appendix I, such as Fig 11.e and 11.f. The previous method has caused considerable damage to other downstream tasks (`-37%`).

---

> > ### Author Response · Authors · 2024-11-29
> >
> > Dear Reviewer dq76,
> >
> > We appreciate your further feedback and have addressed your questions. Meanwhile, Reviewers Waro and Skue have expressed strong recognition of our paper and increased their scores. If you have any additional questions or need further clarification, we welcome your supplementary comments. Thank you for your continued consideration. Happy Thanksgiving and best wishes!
> >
> > Sincerely Yours
> >
> > The Authors

---

> > > ### Comment · Reviewer_dq76 · 2024-11-30
> > > **Official Comment by Reviewer**
> > >
> > > Thanks for the clarifications.
> > >
> > > Now I understand the experiment settings of section 4. It is still ambiguous to me: What is the conclusion of section 4 and how is it related to other sections in your manuscript?

---

> ### Author Response · Authors · 2024-12-01
>
> Thank you for your response. First, section 4 serves as a pivotal link in our paper. It demonstrates the potential of capability neurons for parameter localization, providing early analysis and inspiration for the capability neuron localization discussed in section 5.
>
> In section 3, we show that previous attempts at individual knowledge localization are unreliable. To further analyze the factors that can achieve parameter localization, we construct a decoupled dataset in section 4, which includes a main part and a replaceable part. By using this dataset to locate neurons, we can better control variables and clarify which factors contribute to parameter localization with significant overlap.
>
> As mentioned in the experimental results on line 321, for the same replaceable part—for example, "1 + 1 = ?"—the average overlap of neurons located for 1,000 pairs of subsample 1 (sub1) and subsample 2 (sub2) is only 15.6%. This indicates that individual knowledge localization is unreliable, even when sub1 and sub2 both contain the mathematical problem "1 + 1 = ?".
>
> Interestingly, when we take the intersection of the localization results from 1,000 sub1 samples, the overlap reaches 7.3%. Unlike the previously mentioned 15.6% (which pertains to two samples, sub1 and sub2), this 7.3% represents the overlap among 1,000 samples. This suggests that the model has considerable potential in localizing computational ability (across all sub1) or programming ability (across all sub2).
>
> Furthermore, section 4 reinforces the unreliability of individual knowledge localization. Even when two individual pieces of knowledge share the same replaceable part, we still do not achieve a high overlap. However, 1,000 different samples that reflect the model's computational or programming capabilities involve 7.3% or 8.6% of the same neurons, respectively. This result is both exciting and enlightening. Therefore, in section 5, we further extend this experiment on datasets reflecting mathematical, programming, and linguistic abilities, demonstrating the feasibility of capability neuron localization.
>
> Finally, regarding the experimental results, the 7.3% overlap is less than 15.6% because 7.3% is the overlap of neurons localized across 1,000 samples—a challenging task. In contrast, 15.6% is the average overlap of neurons localized between sub1 and sub2 with the same replaceable part.
>
> To further demonstrate the effectiveness of decoupling experiments, we constructed decoupling data based on the emotion recognition dataset SemEval [1]. For example, the main part is "William did not complete his homework," and the replaceable parts are "Is this sad or joyful?" and "What should he do next."
>
> |Overlap rate|Calculation & programming|Emotion recognition & reasoning|
> |:-:|:-:|:-:|
> |Sub1 & Sub2|15.6|14.3|
> |Sub1 list|7.3|9.7|
> |Sub2 list|8.6|8.9|
>
> We commit to updating this part of the results and explanations in the revised version.
>
> *Ref*
>
> [1] SemEval-2016 Task 5: Aspect Based Sentiment Analysis. Hal.science 2016.

---

> > ### Author Response · Authors · 2024-12-02
> >
> > Dear Reviewer dq76,
> >
> > Thank you for your previous feedback! As the discussion period is coming to an end, do you have any additional questions? We welcome your further inquiries.
> >
> > Sincerely,
> >
> > The Authors

---

> > > ### Author Response · Authors · 2024-12-03
> > >
> > > Dear Reviewer dq76,
> > >
> > > We appreciate your recognition of our paper and the increase in scores! We promise to update the above explanation in the revised version. All the best!
> > >
> > > Sincerely,
> > >
> > > The Authors

---

### Author Response · Authors · 2024-11-19
**Global Response**

We thank the reviewers for their thoughtful feedback. We are glad the reviewers find that:
* Our motivation is innovation and has great significance
    * "The finding is insightful to the community." - dq76
    * "The paper is clear and well motivated." - Skue
* Our approach presents a new perspective on the research question
  * "It clearly showcases the existing methods for understanding LLMs and their limitations." - Skue
  * "The authors propose a change of perspective to address the current limitations of techniques.“ - 4N9w
* Our approach is effective and has a wide range of application in real-world scenarios..
  * "which help the understanding of large-scale neural networks." - dq76
  * "Developed a new method to categorize neurons." - wrao

**[Main Motivation]**

Large-scale language models have achieved superior performance, however, it is still unclear how model parameters affect performance improvement. Despite recent interesting hypotheses suggesting that individual knowledge is stored on local parameters, extensive experimental evidence has shown that previous hypotheses are unreliable. Through decoupling and capability neuron localization experiments, we have demonstrated that capabilities can be localized. Fine-tuning 0.15% of capability neurons will significantly improve model performance, this proposes a new norm of "Localization + Fine-tuning".  As highlighted by reviewer dp76, this work holds considerable significance.

**[Supplementary Experiments]**

* **1. Accuracy experiment.** We performed localization experiments on more settings:

**1.1** A larger model -> Llama2-13B

|ratio|GSM8K|Emotion|Code25K|Meta_Math|Imdb|
|:-:|:-:|:-:|:-:|:-:|:-:|
|overlap|94.26|94.68|91.10|95.37|98.32|
|IoU|88.92|89.79|83.62|91.10|96.68|
|neuron|0.10|0.19|0.11|0.09|0.08|

**1.2** A different model -> GPTj-6B
|ratio|GSM8K|Emotion|Code25K|Meta_Math|Imdb|
|:-:|:-:|:-:|:-:|:-:|:-:|
|overlap|95.66|87.62|91.25|83.62|98.27|
|IoU|91.63|77.96|83.89|71.77|96.60|
|neuron|0.28|0.19|0.11|0.27|0.26|

**1.3** Different $\sigma$ with llama2-7B:

$\sigma = 3$
|ratio|GSM8K|Emotion|Code25K|Meta_Math|Imdb|
|:-:|:-:|:-:|:-:|:-:|:-:|
|overlap|97.21|97.35|89.88|93.47|98.76|
|IoU|94.49|94.84|81.61|87.56|97.55|
|neuron|0.46|0.62|0.30|0.44|0.36|

$\sigma = 12$
|ratio|GSM8K|Emotion|Code25K|Meta_Math|Imdb|
|:-:|:-:|:-:|:-:|:-:|:-:|
|overlap|97.53|98.03|95.49|93.15|97.92|
|IoU|95.16|96.12|91.34|87.10|95.92|
|neuron|0.05|0.06|0.03|0.05|0.04|

* **2. Enhance experiment.** We extended our enhancement experiments (For convenience, we only list the results under the last epoch here.). The best results are in **bold** and `highlighted` means
the suboptimal.

**2.1** A different $\sigma$ with llama2-7B:
$\sigma = 3$
|Method|GSM8K|Emotion|Code25K|$Avg.$|
|:-:|:-:|:-:|:-:|:-:|
|random|23.75|`26.79`|`53.47`|`34.67`|
|w/o located|`25.19`|19.29|42.77|29.08|
|located|**26.31**|**51.62**|**56.02**|**44.65**|

**2.2** A different model -> GPTj-6B:
|Method|GSM8K|Emotion|Code25K|$Avg.$|
|:-:|:-:|:-:|:-:|:-:|
|random|25.69|28.54|`51.07`|35.10|
|w/o located|**32.00**|**38.71**|48.50|`39.73`|
|located|`27.38`|**48.58**|**52.53**|**42.83**|

* **3. Erase experiment.**  We simultaneously perform erase experiments on Llama2-13B:
$\sigma=6$  the specific experimental results are in our updated Table3.

|Dataset|GSM8K|Emotion|Code25K|$Avg.$|
|:-:|:-:|:-:|:-:|:-:|
|Base acc.|0.38|31.96|45.95|26.10|
|random|0.31 ($\downarrow 0.07$)|33.75 ($\uparrow 1.79$)|45.81 ($\downarrow 0.14$)|26.62 ($\uparrow 0.52$)|
|locate|0.00 ($\downarrow 0.38$)|5.38 ($\downarrow 26.58$)|18.06 ($\downarrow 27.89$)|7.81 ($\downarrow 18.29$)|

* **4. Verification experiment.**   We evaluated the image of enhanced neurons on other tasks that are not involved
in training. The following represents the performance improvement of the model in other tasks:

|Model|random|w/o located|located|
|:-:|:-:|:-:|:-:|
|LLama2-7B $\sigma=3$|5.20|-5.58|9.43|
|LLama2-7B $\sigma=6$|3.20|-3.31|11.18|
|GPTJ-6B $\sigma=6$|9.00|14.73|17.66|

We find that more accurate localization methods cause less damage or can even benefit downstream tasks.

* **5. Convergence experiment.** We validated the relationship between different data volumes and model sizes.

Llama2-13B&Emotion[1]:

|Num_data|100|200|300|400|500|600|700|800|900|1000|1100|1200|
|:-:|:-:|:-:|:-:|:-:|:-:|:-:|:-:|:-:|:-:|:-:|:-:|:-:|
|overlap|89.51|93.60|87.56|83.94|95.56|95.37|94.21|95.72|91.76|95.75|94.01|94.68|
|IoU|80.51|87.94|76.35|69.30|91.48|91.11|88.97|91.66|84.22|91.84|88.63|89.79|

GPTj-6B&Emotion[1]:

|Num_data|100|200|300|400|500|600|700|800|900|1000|1100|1200|
|:-:|:-:|:-:|:-:|:-:|:-:|:-:|:-:|:-:|:-:|:-:|:-:|:-:|
|overlap|45.67|79.04|83.85|65.92|77.81|85.85|80.99|77.68|87.82|84.28|91.73|87.62|
|IoU|29.46|65.26|72.18|49.11|63.64|75.20|68.04|63.43|78.26|72.82|84.70|77.96|

**The Rebuttal revision provides more detailed experimental results.**

*Ref*

[1] Context based emotion recognition using emotic dataset. IEEE Trans 2019.

---

### Author Response · Authors · 2024-12-01

Dear Reviewers,

Thank you for your previous suggestions! As the discussion period is coming to an end, do you have any additional questions? We welcome your further inquiries.

Sincerely,

The Authors

---

### Meta-Review · Area_Chair_N7bs · 2024-12-24

**Metareview:**

The paper makes an interesting claim that existing forms of individual knowledge storage are all inaccurate. Instead, the author presents a new framework for localizing “commonality neurons”, the neurons that represent shared attributes or capabilities across datasets. They introduce the Commonality Neuron Locating (CNL) method which identifies neurons that consistently activate for similar tasks or data types. The paper concludes that capabilities can be localized within LLMs, potentially leading to new ways of understanding and manipulating LLM functionality.

The majority (3/4) reviewers rated favorably towards an acceptance, however all marginally. I recommend acceptance for its rather novel insights, and strong rebuttal.

**Additional Comments On Reviewer Discussion:**

It is overall a healthy and productive rebuttal. All reviewers engaged with the author, and 3 of them raised their scores.

---

### Decision · Program_Chairs · 2025-01-22

Accept (Poster)